# Graph-to-Sequence Generation Beyond Autoregressive Models: A Graph-Aware Diffusion Framework

## Abstract

Pre-trained language models (PLMs) remain unreliable for graph-to-sequence (G2S) generation, where two challenges are particularly acute: (i) *factual grounding*, ensuring all entities are faithfully realized, and (ii) *edit sensitivity*, ensuring small, local graph edits to propagate consistently in the output. We propose Diffusion Language Models for Graphs (`DLM4G`), a *non-autoregressive framework* for iterative refinement conditioned on the graph input. Central to `DLM4G` is a graph-aware adaptive noising strategy, where noise is applied to the output sequence aligned with the graph components (entities and relations) using a learnable component-wise schedule. We learn a component-wise schedule by linearly mapping between per-component denoising loss and noise schedule. This ensures entities are generated faithfully and keeps graph edits localized in the text. Through extensive experiments on three benchmark datasets, `DLM4G` outperforms state-of-the-art autoregressive baselines that are 12–127× larger, achieving 10–15% relative gains on standard surface-level metrics (BLEU, ChrF++, METEOR) and embedding-based metrics (BERTScore-F1, MAUVE). More importantly, `DLM4G` improves factual grounding (FGT, ↑) by $+\Delta_{\text{FGT}}$ 4.7 % and edit sensitivity (ESR, ↑) by $+\Delta_{\text{ESR}}$ 7.9 % on average compared to comparably sized autoregressive baselines. Finally, we evaluate `DLM4G` on molecule captioning, where molecular graphs are verbalized into textual descriptions, demonstrating its applicability to biomedical G2S tasks. Our code is available here: CODE

## 1 Introduction

Graphs are a ubiquitous data structure, fundamental to domains like social networks, biological systems, and recommendation platforms (Wang et al., 2021; Fan et al., 2019; Wang et al., 2024b). However, their complex topology makes verbalization difficult. Many downstream tasks such as graph reasoning (Skianis et al., 2024), graph captioning (Hsieh et al., 2025; Li et al., 2024a), graph translation (Xu et al., 2022) require readable, faithful text. To address this challenge, the task of Graph-to-Sequence (G2S) has emerged, which focuses on generating coherent text from graph inputs (Fatemi et al., 2024). Real-world G2S applications include (i) molecular & protein captioning – translating chemical graphs (proteins & molecules) into concise natural-language summaries (Kim et al., 2025) and (ii) Knowledge Graph Question Answering (KGQA)–verbalizing KG subgraphs to support multi-hop reasoning (Wu et al., 2023).

Earlier G2S methods encoded structure explicitly with graph-based encoders (Song et al., 2018; Ribeiro et al., 2019; 2020; Schmitt et al., 2021). Recent work shows that autoregressive pre-trained language models (PLMs) achieve strong performance without *graph-specific inductive biases* on surface-overlap metrics (BLEU, chrF++, METEOR) (Ribeiro et al., 2021). These scores can remain high despite factual omissions and hallucinations. Therefore, these models lack (i) factual grounding (all entities/relations must be realized) and (ii) edit sensitivity (small local graph edits must be reflected predictably). A key factor for these weaknesses in PLMs is left-to-right decoding. This approach leads to early token commitments that reduce sensitivity to local edits, often causing entities or relations to be omitted or misrepresented (Li et al., 2022; Gong et al., 2023). This points to two needs: (1) a modeling choice that preserves global coherence and local faithfulness, ensuring while

reflecting small edits and realizing all entities/relations, and (2) an evaluation criterion that directly measures grounding and edit sensitivity.

To address these, we propose a two-pronged approach: *First:* regarding the modeling choice, we investigate non-autoregressive (NAR) diffusion-based language models for G2S. Here, we navigate a key design trade-off: while graph-specific encoders (e.g., GNNs) offer strict structural guarantees like permutation equivariance ("hard" inductive bias), they often lack the semantic richness of PLMs. We therefore use a PLM backbone for `DLM4G` but re-introduce structure via a "soft" inductive bias— graph-aware noising schedule. This mechanism is critical because, while diffusion models generally support self-correction via iterative denoising (Sahoo et al., 2024; Chuang et al., 2024; Yuan et al., 2024; Gong et al., 2025; Venkatraman et al., 2025), standard data-agnostic schedules corrupt factual and syntactic elements equally (Ho et al., 2020). This uniform corruption undermines factual grounding. `DLM4G` overcomes this by using the graph-aware schedule to strategically preserve factual information during the noising process. *Second*, to address the evaluation gap, we introduce simple, task-grounded metrics that move beyond surface-level overlap to directly measure factual grounding and edit sensitivity.

To sum up, our overall contributions: (1) A novel, graph-aware noising schedule to improve factual grounding; (2) State-of-the-art performance on three diverse datasets across a wide range of metrics; (3) Two new task-grounded metrics to evaluate factual grounding and edit sensitivity; and finally (4) An extension of our framework to the real-world scientific task of molecule captioning.

## 2 BACKGROUND AND PRELIMINARIES

This section first reviews related work in graph-to-sequence generation. Then, to ground our contributions, we present the preliminary concepts of standard denoising diffusion models. We briefly review relevant work, deferring full technical details to Appendix A.2.

### 2.1 RELATED WORK

**Graph-to-Sequence Learning**: G2S has progressed from (i) template-based systems Wiseman et al. (2018); Kasner & Dusek (2022); Vejvar & Fujimoto (2023), to (ii) neural encoder–decoders with learned graph embeddings Wiseman et al. (2017); Beck et al. (2018); Iso et al. (2019), and (iii) fine-tuned transformers achieving state-of-the-art fluency and factuality Vaswani et al. (2023); Ribeiro et al. (2021). This evolution frames the current G2S landscape.

**PLMs for Graphs**: Leveraging LLMs for graph verbalisation involves following challenges: (i) *alignment* of graph elements to words Zhu et al. (2025), and (ii) *multi-level semantics* across nodes, edges, and subgraphs Wang et al. (2024a). This taxonomy spans Graph-to-Sequence (G2S) to Graph-to-Token (G2T). KG-to-text models use positional encodings, prompts, and multi-granularity attention Zhu et al. (2025), reducing omissions but still constrained by left-to-right decoding. Diffusion LMs, with iterative denoising, could overcome these limitations.
Prior PLM-based G2S work treats the input KG as a serialized sequence of relational triples, using special markers `[HEAD]`, `[REL]`, `[TAIL]`, and `[SEP]` (see Section 4.1). This design allows us to plug into standard encoder–decoder Transformers while giving up strict permutation invariance, a trade-off we revisit in Limitations 5.

**Diffusion Models for Conditional Generation**: Conditional diffusion guides denoising with an input sequence encoding, extending conditional-VAE ideas Zhao et al. (2017). Early text models (Diffusion-LM Li et al. (2022), Analog Bits Chen et al. (2023)) imposed weak conditioning via classifiers or plug-in controls, while DIFFUSEQ Gong et al. (2023) enabled true sequence-to-sequence conditioning in continuous space. `DLM4G` builds on this foundation and combines classifier-free diffusion with explicit KG conditioning as the control variable for more coherent KG verbalisation.

**Molecule Captioning**: Prior AR/NAR captioning approaches for molecules inherit these limitations Edwards et al. (2022); Liu et al. (2024a). Table 8 compares these paradigms with `DLM4G`.

### 2.2 PRELIMINARIES: DENOISING DIFFUSION MODELS

Denoising diffusion probabilistic models (DDPMs) are generative models that learn a data distribution, often conditioned on some context $\mathbf{c}$, $p(\mathbf{z}_0 \mid \mathbf{c})$. They consist of a fixed forward process and a learned reverse process.

**Forward process**: A standard DDPM forward process corrupts clean data $\mathbf{z}_0$ through a Markov chain with noise-schedule coefficients $\{\alpha_t\}_{t=1}^{T}$ controlling signal decay. This yields the standard closed-form for sampling a noised state $\mathbf{z}_t$ at any timestep $t$:

$$\mathbf{z}_t = \sqrt{\bar{\alpha}_t}\,\mathbf{z}_0 + \sqrt{1 - \bar{\alpha}_t}\,\boldsymbol{\epsilon} \quad \text{with} \quad \bar{\alpha}_t = \prod_{s=1}^{t} \alpha_s \text{ and } \boldsymbol{\epsilon} \sim \mathcal{N}(\mathbf{0}, \mathbf{I}). \tag{1}$$

Standard diffusion models typically use a fixed, data-agnostic (isotropic) noise schedule.

**Reverse process with Conditional Denoising**: The reverse process learns to recover the clean data $\mathbf{z}_0$ from pure noise $\mathbf{z}_T \sim \mathcal{N}(\mathbf{0}, \mathbf{I})$. It is defined as a Markov chain $p_\theta(\mathbf{z}_{0:T})$ where each reverse transition $p_\theta(\mathbf{z}_{t-1} \mid \mathbf{z}_t, \mathbf{c})$ is a Gaussian whose mean $\boldsymbol{\mu}_\theta$ and variance $\boldsymbol{\Sigma}_\theta$ are parameterized by a model $\mathcal{M}_\theta(\mathbf{z}_t, t, \mathbf{c})$. The model is trained to predict the mean of the true posterior $q(\mathbf{z}_{t-1} \mid \mathbf{z}_t, \mathbf{z}_0)$. The model parameters $\theta$ are optimized by maximizing the variational lower bound (VLB) on the conditional log-likelihood:

$$\mathcal{L}_{\text{vlb}} = \mathbb{E}_q\Big[\underbrace{-\log p_\theta(\mathbf{z}_0|\mathbf{z}_1, \mathbf{c})}_{\text{Reconstruction } (L_0)} + \sum_{t=2}^{T} \underbrace{D_{KL}\big(q(\mathbf{z}_{t-1}|\mathbf{z}_t, \mathbf{z}_0)||p_\theta(\mathbf{z}_{t-1}|\mathbf{z}_t, \mathbf{c})\big)}_{\text{Denoising Matching } (L_{t-1})} + \underbrace{D_{KL}\big(q(\mathbf{z}_T|\mathbf{z}_0)||p(\mathbf{z}_T)\big)}_{\text{Prior Matching } (L_T)}\Big] \tag{2}$$

While tractable, direct optimization of the full VLB is often unstable.

## 3 THE DLM4G METHODOLOGY

### 3.1 PROBLEM STATEMENT

Let $\mathcal{G} = (\mathbf{V}, \mathbf{E}, \mathbf{X}, \mathcal{R})$ be the input graph, where $\mathbf{V} = \{v_1, \dots, v_n\}$ is the set of nodes, $\mathbf{X} = \{x_1, \dots, x_n\}$, with each $x_i \in \mathbb{R}^d$, representing the associated node features, and $\mathbf{E} \subseteq \mathbf{V} \times \mathcal{R} \times \mathbf{V}$ denotes a set of directed edges representing relations $r_{ij} \in \mathcal{R}$. In many settings, such as KGs, each relation type $r_{ij} \in \mathcal{R}$ is associated with a feature vector $f_r \in \mathbb{R}^k$, capturing its semantic properties. This structure can be expressed as a sequence of relational triplets $\tilde{\mathcal{G}} = \{(h_i, r_{ij}, t_j)\}_{\substack{i,j=1 \\ i \neq j}}^{n}$, where $h_i, t_j \in \mathbf{V}$ are head and tail entities, respectively, and $r_{ij} \in \mathcal{R}$ is the relation type. The goal is to learn a model that maps such structured graph inputs to meaningful output sequences. Formally, a parameterized DLM4G model $\mathcal{M}_\theta$ is trained to predict the corresponding output sequence:

$$\mathcal{M}_\theta : \tilde{\mathcal{G}} \to \mathbf{S}, \tag{3}$$

where $\mathbf{S} = \{s_i \in \mathcal{W} \mid 1 \leq i \leq \mathrm{N}\}$ is a sequence of fixed length N, and $\mathcal{W}$ denotes the target vocabulary. Formally, we aim to learn this conditional distribution $p(\mathbf{S} \mid \tilde{\mathcal{G}}; \theta)$, that approximates the underlying data distribution. To achieve this, we introduce DLM4G, a novel diffusion framework.

### 3.2 THE DLM4G FRAMEWORK OVERVIEW

We introduce DLM4G (framework is shown in Figure 1), a denoising diffusion framework designed to generate factually-grounded text from KGs. We adapt the standard conditional DDPM framework (reviewed in Section 2.2) to the graph-to-sequence task, where we learn the distribution $p(\mathbf{S} \mid \tilde{\mathcal{G}})$. To apply the diffusion process to discrete text, we first define our "clean data" $\mathbf{z}_0$ as the continuous representation of the sequence $\mathbf{S}$, obtained via a learnable embedding layer: $\mathbf{z}_0 = g_\Phi(\mathbf{S}) \in \mathbb{R}^{N \times d}$. The "conditioning context" $\mathbf{c}$ (from the preliminaries) is our input graph, $\tilde{\mathcal{G}}$. Following the PLM-based G2S convention introduced in Section 2, we realize $\tilde{\mathcal{G}}$ as a token sequence $\tilde{\mathcal{G}} = \langle \texttt{[HEAD]}\ h_i\ \texttt{[REL]}\ r_{ij}\ \texttt{[TAIL]}\ t_j \rangle$ with $\texttt{[SEP]}$ delimiters, which is fed to $\mathcal{M}_\theta$ as the conditioning signal (see Section 4.1). Our reverse-process is therefore $\mathcal{M}_\theta(\mathbf{z}_t, t, \tilde{\mathcal{G}})$.

The core of our approach differs from standard diffusion in two key aspects:

1. **A Graph-Aware Noising Schedule:** Instead of the isotropic schedule (Eq. 1), our forward process uses a graph-aware adaptive noising schedule. This is detailed in Section 3.3.

2. **A Simplified Training Objective:** Instead of the full VLB (Eq. 2), we use a simplified objective (detailed in Sec. 3.4) that trains $\mathcal{M}_\theta$ to directly predict $\mathbf{z}_0$.

## 3.3 GRAPH AWARE NOISING SCHEDULE

**Motivation & Rationale**: Standard diffusion models rely on fixed, data-agnostic noising schedules that apply noise uniformly across all tokens (Ho et al., 2020; Nichol & Dhariwal, 2021). This is suboptimal for graph-to-text generation, as it corrupts critical factual tokens (entities, relations) and simple syntactic tokens equally (Yuan et al., 2024). Since recovering factual content from noisy states is relatively difficult, applying noise uniformly weakens factual grounding. To address this, we introduce a *graph-aware* noising schedule. The core idea is to use the model's per-timestep reconstruction error for a graph-aligned token, $\ell_t^i$, as a proxy for its difficulty with factual consistency. By building a mapping from this empirical difficulty to the noise schedule (details in Stage 2), we re-parameterize the denoising path as a function of prediction error. This creates a more stable trajectory for factual content, thus improving factual grounding and sensitivity to graph edits.

**Graph–sequence alignment (Training)**: To enable our graph-aware noising schedule, we first perform a one-time offline alignment to map tokens in the target sequence ($\mathbf{S}$) to their corresponding entities and relations in the graph ($\tilde{\mathcal{G}}$). The pipeline operates in three stages: (i) generating all possible names and aliases for each entity; (ii) detecting mentions of these names in the text using a NER model (Zaratiana et al., 2024); and (iii) linking these mentions to the correct graph entity to resolve ambiguities (Xin et al., 2024; Liu et al., 2024b; Ding et al., 2024). The result is an alignment map $\mathcal{A}$ connecting token indices in $\mathbf{S}$ to graph elements, which is used exclusively during training. A detailed analysis of the alignment module is in Appendix A.4.

**Noising Schedule**: We apply graph-aware noising for the graph–text aligned set $\mathcal{A}$, while keeping unaligned tokens on the baseline *sqrt* schedule. The procedure has two stages, summarized in Alg 1.

**Stage 1: Estimating token-wise difficulty.** For each aligned token $i \in \mathcal{A}$ and diffusion step $t = 1, \ldots, T$, we define the denoising difficulty as:

$$\ell_t^i = \mathbb{E}_{\mathbf{z}_t \sim q(\mathbf{z}_t | \mathbf{z}_0)} \left\| \mathcal{M}_\theta(\mathbf{z}_t, t, \tilde{\mathcal{G}})^{(i)} - \mathbf{z}_0^{(i)} \right\|^2 \quad (4)$$

Averaging over the training set yields a difficulty profile $(\ell_1^i, \ldots, \ell_T^i)$ for each $i \in \mathcal{A}$. Empirically, $\ell_t^i$ tends to increase with $t$ (later steps are noisier), but the estimated profile is not strictly monotone. We also compute $\ell_{\min}^i = \min_t \ell_t^i$ and $\ell_{\max}^i = \max_t \ell_t^i$ to define the difficulty range for token $i$. In Stage 2 these profiles and their ranges are used to construct a token-wise cumulative schedule, and to obtain a monotonic difficulty profile for each token $i$.

**Stage 2: Token-wise schedule.** Given $(\ell_t^i)_{t=1}^T$ and the baseline cumulative schedule $(\bar{\alpha}_t)_{t=1}^T$, we construct an adaptive schedule $(\bar{\alpha}_{t,\text{new}}^i)_{t=1}^T$ for each $i \in \mathcal{A}$. Since this schedule controls noise applied at each step, we want to reallocate noise according to denoising difficulty. Hence, we define a piecewise-linear map $\Psi_i : [\ell_{\min}^i, \ell_{\max}^i] \to (0, 1)$ that interpolates the baseline schedule as a function of loss:

---

**Algorithm 1** Graph-Aware Adaptive Noising

**Require:** Baseline cumulative schedule $\{\bar{\alpha}_t\}_{t=1}^T$, alignment set $\mathcal{A}$, update interval $\mathtt{K}$
**Ensure:** Schedules $\{\bar{\alpha}_{t,\text{new}}^i\}_{t=1}^T$ for $i \in \mathcal{A}$
1: **if** $\mathtt{train\_step} \% \mathtt{K} == 0$ **then**
2:    **for** $i \in \mathcal{A}$ **do**
3:       Estimate $\{\ell_t^i\}_{t=1}^T$ via Eq. 4; compute $\{\ell_{\min}^i, \ell_{\max}^i\}$.
4:       Define piecewise-linear map $\Psi_i$ as in Eq. (5).
5:       Construct difficulty ramp $\{\ell_t^{i,\text{new}}\}_{t=1}^T$ via Eq. (6).
6:       Compute new schedule $\tilde{\alpha}_t^i = \Psi_i(\ell_t^{i,\text{new}})$.
7:       Clamp $\tilde{\alpha}_t^i$ to $(0, 1)$, apply a non-increasing isotonic projection to obtain $\{\bar{\alpha}_{t,\text{new}}^i\}_{t=1}^T$.
8:    **end for**
9: **end if**
10: For any $i \notin \mathcal{A}$, set $\bar{\alpha}_{t,\text{new}}^i = \bar{\alpha}_t$.
11: Compute per-step coefficients $\alpha_{t,i} = \bar{\alpha}_{t,\text{new}}^i / \bar{\alpha}_{t-1,\text{new}}^i$, $\beta_{t,i} = 1 - \alpha_{t,i}$, with $\bar{\alpha}_{0,\text{new}}^i = 1$.
12: **return** $\{\bar{\alpha}_{t,\text{new}}^i\}$ for all tokens $i$ and steps $t$.

---

$$\Psi_i(x) = \bar{\alpha}_{t-1} + \frac{\bar{\alpha}_t - \bar{\alpha}_{t-1}}{\ell_t^i - \ell_{t-1}^i} \left(x - \ell_{t-1}^i\right), \qquad x \in [\ell_{t-1}^i, \ell_t^i), \ \ t = 2, \ldots, T, \quad (5)$$

with $\Psi_i(\ell_1^i) = \bar{\alpha}_1$ and $\Psi_i(\ell_T^i) = \bar{\alpha}_T$. In case $\ell_t^i = \ell_{t-1}^i$, we add a tiny jitter $\varepsilon$ to avoid division by zero. Empirically, $(\ell_t^i)_{t=1}^T$ is not strictly monotone in $t$, so instead of using its raw values we introduce a new linear ramp in difficulty space:

$$\ell_t^{i,\text{new}} = \ell_{\min}^i + \frac{t-1}{T-1} \left(\ell_{\max}^i - \ell_{\min}^i\right), \qquad t = 1, \ldots, T. \quad (6)$$

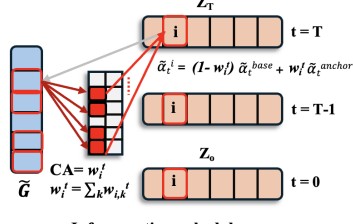

**(A) Graph representation, Alignment**   **(B) Training with Graph-aware Noising**   **(C) Inference**

Figure 1: `DLM4G` framework: (A) Graph-Sequence alignment set $\{\mathcal{A}\}$, obtains the aligned tokens; (B) The model is trained with a graph-aware noising schedule (C) Trained `DLM4G` samples output sequence conditioned on graph.

Substituting this $\ell_t^{i,\text{new}}$ into $\Psi_i(x)$ we get a new cumulative schedule $\tilde{\alpha}_t^i = \Psi_i(\ell_t^{i,\text{new}})$ for $t = 1, \ldots, T$. We clamp $\tilde{\alpha}_t^i$ to $(0, 1)$ and apply a non-increasing isotonic projection (refer Appendix A.5) over $t$ to obtain the final schedule $0 < \bar{\alpha}_{t+1,\text{new}}^i \leq \bar{\alpha}_{t,\text{new}}^i < 1$ for all $t$. We get the forward coefficients for aligned tokens as $\alpha_{t,i} = \bar{\alpha}_{t,\text{new}}^i / \bar{\alpha}_{t-1,\text{new}}^i$, with $\bar{\alpha}_{0,\text{new}}^i = 1$. For unaligned tokens $i \notin \mathcal{A}$, we keep the baseline schedule: $\bar{\alpha}_{t,\text{new}}^i = \bar{\alpha}_t$, hence $\alpha_{t,i} = \alpha_t$.

### 3.4 MODEL TRAINING AND INFERENCE

**Training**: Our training objective is derived from the Variational Lower Bound (VLB) (Eq. 2) presented in the preliminaries. While the full VLB optimization can be unstable (Ho et al., 2020), a common simplification is to train the model $\mathcal{M}_\theta$ to predict the added noise $\epsilon$. However, our framework adopts an alternative $\mathbf{z}_0$-prediction reparameterization, which trains the model to directly predict the clean data $\mathbf{z}_0$ at every timestep $t$. A critical component of this objective is the rounding term $L_0 = -\log \tilde{p}_\Phi(\mathbf{S} \mid \mathbf{z}_0)$, which handles the final step of converting the continuous latent variable $\mathbf{z}_0$ back into discrete tokens $\mathbf{S}$. We define this as a trainable rounding distribution: $\tilde{p}_\Phi(\mathbf{S} \mid \mathbf{z}_0) = \prod_{i=1}^{N} \tilde{p}_\Phi(s_i \mid \mathbf{z}_{0,i})$, where each token $s_i$ is sampled from a softmax distribution over the vocabulary, using logits derived from the corresponding output embedding $\mathbf{z}_{0,i}$. By combining this rounding term (for $t = 0$) with the denoising matching terms (for $t > 1$) using our $\mathbf{z}_0$-prediction reparameterization, we arrive at our final, composite objective. (The full derivation from the VLB is in App A.1).

$$\mathcal{L}_{\text{e2e-simple}}(\mathbf{S}) = \mathbb{E}_q\Big[\sum_{t=2}^{T} \underbrace{\|\mathcal{M}_\theta(\mathbf{z}_t, t, \tilde{\mathcal{G}}) - \mathbf{z}_0\|^2}_{\text{Denoising}} + \underbrace{\| g_\Phi(\mathbf{S}) - \mathcal{M}_\theta(\mathbf{z}_1, 1, \tilde{\mathcal{G}})\|^2}_{\text{Consistency}} - \underbrace{\log \tilde{p}_\Phi(\mathbf{S} \mid \mathbf{z}_0)}_{\text{Rounding}}\Big] \quad (7)$$

This objective directly optimizes the most critical parts of the process: the denoising accuracy across all steps (Denoising), the consistency of the first denoising step with the true data embedding (Consistency), and the quality of the final conversion to discrete tokens (Rounding).

**Inference-time schedule.** At test time the alignment set $\mathcal{A}$ is unavailable, so we use cross-attention as a proxy. We blend the baseline cumulative schedule $\bar{\alpha}_t^{\text{base}}$ with a fixed "anchor" schedule $\bar{\alpha}_t^{\text{anchor}}$, obtained by averaging the aligned-token schedules learned during training. For each decoder token $i$ at denoising step $t$, we compute a scalar weight $w_i^t \in [0, 1]$. This weight is calculated by summing the normalized cross-attention weights, $w_{i,k}^t$, from the **final decoder layer's** cross-attention module. The sum is taken over the encoder positions corresponding to the serialized graph $\tilde{\mathcal{G}}$: $w_i^t = \sum_{k \in \tilde{\mathcal{G}}} w_{i,k}^t$, where $w_{i,k}^t$ denotes the normalized cross-attention weight from decoder token $i$ to encoder token $k$.

Figure 2: Inference-time schedule. Cross-attention (**CA**) from decoder token $i$ to graph tokens in $\tilde{\mathcal{G}}$ is aggregated into $w_i^t$, which blends the baseline and anchor schedules for each token $i$.

The resulting per-token cumulative schedule is: $\tilde{\alpha}_t^i = (1 - w_i^t)\,\bar{\alpha}_t^{\text{base}} + w_i^t\,\bar{\alpha}_t^{\text{anchor}}$. Tokens that attend strongly to graph tokens in $\tilde{\mathcal{G}}$ follow the "anchor" schedule, while purely syntactic tokens follow the baseline schedule, serving as a lightweight proxy for the explicit alignment used during training.

## 4 EXPERIMENTS

### 4.1 EXPERIMENTAL SETUP

**Model Architecture**: DLM4G is an encoder–decoder Transformer that conditions on the serialized KG input (see *graph representation* §4.1). We evaluate two variants: a 6-encoder/6-decoder configuration ($\approx$ 50M parameters; DLM4G-1.o) and a 6-encoder/9-decoder configuration ($\approx$ 63M; DLM4G-2.o), both using GeLU activations Vaswani et al. (2023); Hendrycks & Gimpel (2023). Inputs are tokenized with the bert-base-uncased vocabulary Devlin et al. (2019); the control tokens [HEAD], [REL], [TAIL], and [SEP] are introduced as learned special tokens with dedicated embeddings. Other components follow the standard Transformer encoder–decoder design.

**Graph Representation**: We represent the set of relational triples $\tilde{\mathcal{G}}$, as a single linearized sequence. This is achieved by serializing each triplet $(h_i, r_{ij}, t_j) \in \tilde{\mathcal{G}}$ into a string format using special tokens, e.g: $\langle$[HEAD] $h_i$ [REL] $r_{ij}$ [TAIL] $t_j\rangle$, and concatenating them with a separator token [SEP]. For all KG benchmarks (WikiOFGraph, GenWiki, TekGEN), we simply preserve the triple order provided in the released datasets and do not reorder or subsample triples; this dataset-defined order serves as our consistent traversal for linearization. We adopt linearization for the following reasons: (i) it plugs into off-the-shelf backbones and decoding stacks, making ablations across baselines directly comparable; (ii) Transformer self-attention can model long-range interactions across the flattened triples, which is important for faithful realization; and (iii) prior work shows strong performance for linearized KG→text with PLMs, even without graph-specific inductive bias (Ribeiro et al., 2021; Wang et al., 2024a). Example *(graph→sequence)*:

Serialized KG ($\tilde{\mathcal{G}}$): $\langle$[HEAD] USA [REL] hosted [TAIL] 1994_FIFA_World_Cup$\rangle$
[SEP] $\langle$[HEAD] USA [REL] capital [TAIL] Washington_D.C.$\rangle$
[SEP] $\langle$[HEAD] 1994_FIFA_World_Cup [REL] top_scorer [TAIL]
Hristo_Stoichkov$\rangle$.
Corresponding sequence (**S**): *"The United States hosted the 1994 FIFA World Cup; its capital is Washington, D.C., and the tournament's top scorer was Hristo Stoichkov"*.

**Datasets**: We use three datasets for our experiments: (1) *WikiOFGraph* (Kim et al., 2024), a 5.85M-sample dataset ontology-free dataset for graph-text task; (2) *GenWiki* (Jin et al., 2020), an unsupervised dataset of 680K Wikipedia text and DBpedia graph pairs, with a focus on entity overlap and a 1K human-annotated test set; and (3) *TekGEN* (Agarwal et al., 2021), a dataset of 6.3M sentences generated by verbalizing Wikidata triples. More details are available in Appendix A.3.

**Baselines**: We benchmark DLM4G, against four categories of baselines:
(i) *Pretrained-LM baselines*, comprising finetuned GPT-2 (Small/Base) (Mager et al., 2020), and T5 (Small/Large) (Ribeiro et al., 2021) on all datasets;
(ii) *Zero-shot evaluation*, deploying GPT-o4-mini (8 B), LLaMa-3-8B (8 B), Qwen 2.5 (7 B) and DeepSeek (7 B) to assess off-the-shelf generalization without any task-specific finetuning and
(iii) SOTA G2S methods, including ReGen on TekGen (Dognin et al., 2021) and the Ontology-Free (Kim et al., 2024), Rule-Based (Schmitt et al., 2020), and Direct-Transfer, Noisy-Supervised (Koncel-Kedziorski et al., 2019) baselines on WikiofGraph and GenWiki (excluding CycleGT$_{Base}$ due to non-standard splits in prior work (Jin et al., 2020; Guo et al., 2020)).
(iv) *Diffusion baselines*, including Diffuseq Gong et al. (2023), FlowSeq Hu et al. (2024) and SeqDiffuSeq Yuan et al. (2024) adapted specifically to the G2S task.

**Implementation Details and Evaluation Metrics**: We train DLM4G with diffusion process of $T = 2000$ timesteps, using our graph-aware noising schedule, and inputs are tokenized using the bert-base-uncased vocabulary (Devlin et al., 2019). Training uses a peak learning rate of $10^{-4}$, 10,000 warm-up steps, and a linear decay schedule, with the adaptive noising schedule updated every 20,000 steps. Full implementation details are provided in Appendix A.6 A.8. For evaluation, we report BLEU (**B**) (Papineni et al., 2002); chrF++ (**CrF++**) (Popović, 2015); and METEOR (**M**) (Banerjee & Lavie, 2005). In addition, we include MAUVE (**MVE**) (Pillutla et al., 2023) for distributional similarity and BERTScore-F1 (**B-F1**) (Zhang et al., 2020) as an embedding-based semantic similarity metric.
Beyond these, we introduce two task-grounded metrics: Factual Grounding Metric (**FGT**), which emphasizes recall by checking that all entities present in the input graph are faithfully realized in the text, and Edit Sensitivity Rate (**ESR**), which emphasizes precision by testing that small, local edits to the graph propagate consistently-i.e. the output highlights only the modifications.

## 4.2 EXPERIMENTAL RESULTS

We evaluate our design using four different methods: (1) full fine-tuning, (2) zero-shot prompting, (3) state-of-the-art (SOTA) benchmarking and (4) Diffusion baselines. Throughout these tests, we carefully balance model size (**#P**arameters) with the amount of data (graph-to-sequence pairs).

For full fine-tuning, we train large models on a dataset of 100,000 graph-to-sequence pairs and test them on a separate set of 1,000 graphs. In the zero-shot evaluation, we use state-of-the-art LLMs without providing any specific training examples. The results across different performance metrics are shown in Table 1. We compare these outcomes against our own pre-trained DLM4G family of small models (approx. 50-63M parameters). These models, trained on an 80/10/10 split, are evaluated on the same test set. A separate SOTA benchmarking table (see Section 4.2) compares DLM4G's performance against other task-specific models.

Table 1: Performance of `DLM4G` compared with (i) finetuning and (ii) zero-shot evaluation paradigms.

| Model | #P | WikiOFGraph | | | GenWiki | | | TekGEN | | |
|---|---|---|---|---|---|---|---|---|---|---|
| | | B | CrF++ | M | B | CrF++ | M | B | CrF++ | M |
| *# Pretrain* | | | | | | | | | | |
| DLM4G-1.o | 50M | 0.619 | 0.823 | 0.688 | 0.401 | 0.663 | 0.527 | 0.247 | 0.493 | 0.375 |
| DLM4G-2.o | 63M | **0.654** | **0.844** | **0.791** | **0.469** | **0.748** | **0.574** | **0.253** | **0.522** | **0.414** |
| %Gain | x1.3↓ | +5.7% | +2.5% | +14.9% | +16.9% | +12.8% | +8.9% | +2.4% | +5.9% | +10.4% |
| *# Finetune* | | | | | | | | | | |
| GPT-2 (S) | 124M | 0.166 | 0.428 | 0.487 | 0.280 | 0.465 | 0.435 | 0.226 | 0.358 | 0.208 |
| GPT-2 (B) | 355M | 0.285 | 0.572 | 0.490 | 0.312 | 0.470 | 0.425 | 0.228 | 0.366 | 0.211 |
| T5 (S) | 60M | 0.385 | 0.688 | 0.471 | 0.227 | 0.495 | 0.447 | 0.189 | 0.352 | 0.203 |
| T5 (L) | 770M | **0.658** | 0.807 | 0.516 | 0.361 | 0.567 | 0.338 | 0.199 | 0.370 | 0.211 |
| DLM4G-2.o | 63M | 0.654 | **0.844** | **0.791** | **0.469** | **0.748** | **0.574** | **0.253** | **0.522** | **0.414** |
| %Gain | x12↑ | 0.0% | +4.5% | +53.3% | +29.9% | +31.9% | +28.4% | +10.9% | +41.1% | +96.2% |
| *# Zero-shot* | | | | | | | | | | |
| LLaMa-3 | 8B | 0.622 | 0.801 | 0.781 | 0.461 | 0.709 | 0.510 | 0.176 | 0.341 | 0.251 |
| Qwen2.5 | 7B | 0.622 | 0.681 | 0.743 | 0.461 | 0.697 | 0.501 | 0.182 | 0.312 | 0.234 |
| DeepSeek | 7B | 0.633 | 0.809 | 0.752 | 0.391 | 0.688 | 0.533 | 0.121 | 0.345 | 0.256 |
| GPT-o4-mini | 8B | 0.648 | 0.847 | 0.783 | 0.464 | 0.734 | 0.471 | 0.121 | 0.327 | 0.277 |
| DLM4G-2.o | 63M | **0.654** | 0.844 | **0.791** | **0.469** | **0.748** | **0.574** | **0.253** | **0.522** | **0.414** |
| %Gain | x127↑ | 0.0% | 0.0% | +1.0% | +1.1% | +2.1% | +7.7% | +39.0% | +51.3% | +49.5% |

**Model Development and Scaling**: We started by pre-training the `DLM4G` family. The `DLM4G-2.o` (63 M #P) model was the best performer across all three datasets. Increasing the model size by a modest 1.3x (from 50M to 63M parameters) resulted in a significant performance boost of 2.4% to 16.9%. This suggests that further scaling `DLM4G` is a promising direction.

**Performance Against Large-Scale Models**: Using our best model, `DLM4G-2.o`, we then benchmarked it against competitors that are 10 to over 100 times larger. In full fine-tuning tests against baselines like the 770M parameter T5-Large, our model performed better on nearly every metric, posting gains up to 96.2%. Furthermore, in zero-shot comparisons against models approximately 127x larger (including LLaMa-3 and GPT-o4-mini), `DLM4G-2.o` remained highly competitive and notably outperformed all of them on the TeKGen dataset. The results are in Table 1

**Semantic Evaluation**: To move beyond traditional surface-level metrics and gain a deeper semantic understanding, we also performed experiments using embedding-based metrics. For this analysis, we compare our model against the best-performing autoregressive baselines using the MAUVE score and BERTScore F1. The results of this comparison are detailed in Table 2.

Table 2: `DLM4G` across embedding based metrics.

| Dataset | Metric | T5 (L) *# Finetune* | GPT-o4-mini *# Zero-shot* | DLM4G-2.o *# Pretrain* | %Gain |
|---|---|---|---|---|---|
| WikiOFGraph | MVE | 0.980 | **0.983** | 0.981 | +0.0% |
| | B-F1 | 0.926 | 0.960 | **0.963** | +0.0% |
| GenWiki | MVE | 0.852 | 0.811 | **0.892** | +4.7% |
| | B-F1 | 0.812 | 0.865 | **0.899** | +3.9% |
| TekGEN | MVE | 0.803 | 0.751 | **0.820** | +2.1% |
| | B-F1 | 0.789 | 0.652 | **0.847** | +7.3% |

**Analysis of Results:** Table 2 shows that `DLM4G-2.o` achieves a SoTA performance on the GenWiki and TekGEN datasets. The most significant improvements are on the TekGEN dataset, where our model shows a +7.3% gain in BERTScore F1 over the next best model. Similarly, on GenWiki, `DLM4G-2.o` improves the SOTA by +4.7% on the MAUVE score. On the WikiOF-Graph, our model achieves the highest BERTScore F1.

These results demonstrate that `DLM4G`-2.o, as a compact pre-trained model, generates semantically rich output that moves beyond simple n-gram matching metrics.

**Primary Finding**: A key takeaway from these results is that a graph-aware pre-training strategy can enable compact models to match, or even surpass, the performance of much larger task-specific and general-purpose LLMs. Finally, to complete our evaluation, we benchmark `DLM4G` against other state-of-the-art (SOTA) models designed specifically for this task.

**SoTA Benchmarking**: The results in the Table 3 confirm that `DLM4G`-2.o consistently outperforms specialized baselines. On the TekGEN dataset, our model establishes a new SOTA on all five metrics, with performance gains reaching as high as +96.2% on METEOR. The results are similarly strong on GenWiki, where `DLM4G`-2.o sets a new SOTA on four of the five metrics and nearly matching the baseline's performance on the final one. Its robust performance across both surface-level and embedding-based metrics highlights the model's ability to generate text that is both lexically accurate and semantically coherent.

**Diffusion Baselines**: Finally, we evaluate `DLM4G` against other diffusion-based text generation models (Table 4). Despite being nearly $1.5\times$ smaller than the strongest baseline (91M vs. 63M), `DLM4G`-2.o demonstrates superior efficiency, consistently outperforming all baselines across every dataset and metric.

Table 3: Performance of `DLM4G` compared with baselines on (a) GenWiki and (b) TekGEN.

| | **GenWiki** | | | | | | **TekGEN** | | | | |
|---|---|---|---|---|---|---|---|---|---|---|---|
| **Baselines** | **B** | **CrF++** | **M** | **B-F1** | **MVE** | **Baselines** | **B** | **CrF++** | **M** | **B-F1** | **MVE** |
| Rule-Based | 0.219 | 0.360 | 0.397 | 0.679 | 0.822 | Rule-based | 0.189 | 0.309 | 0.301 | 0.509 | 0.672 |
| Direct-Transfer | 0.234 | 0.483 | 0.332 | 0.808 | 0.801 | ReGen-SCST | 0.219 | 0.385 | 0.223 | 0.698 | 0.719 |
| Noisy-Sup. | 0.384 | 0.623 | 0.414 | 0.878 | **0.901** | ReGen-CE | 0.199 | 0.372 | 0.214 | 0.612 | 0.701 |
| DLM4G-1.o | 0.401 | 0.663 | 0.527 | 0.857 | 0.841 | DLM4G-1.o | 0.247 | 0.493 | 0.375 | 0.795 | 0.781 |
| DLM4G-2.o | **0.469** | **0.748** | **0.574** | **0.899** | 0.892 | DLM4G-2.o | **0.253** | **0.522** | **0.414** | **0.847** | **0.820** |
| %Gain | +22.1% | +20.0% | +38.6% | +2.4% | 0.0% | %Gain | +10.9% | +41.1% | +96.2% | +21.3% | +14.0% |

Table 4: Performance of `DLM4G` compared with diffusion baselines.

| **Model** | **#P** | **WikiOFGraph** | | | | **GenWiki** | | | | **TekGEN** | | | |
|---|---|---|---|---|---|---|---|---|---|---|---|---|---|
| | | **B** | **M** | **B-F1** | **MVE** | **B** | **M** | **B-F1** | **MVE** | **B** | **M** | **B-F1** | **MVE** |
| *# Diffusion* | | | | | | | | | | | | | |
| FlowSeq | 91M | 0.488 | 0.508 | 0.901 | 0.830 | 0.133 | 0.387 | 0.855 | 0.672 | 0.091 | 0.223 | 0.673 | 0.409 |
| DiffuSeq | 91M | 0.628 | 0.619 | 0.923 | 0.942 | 0.432 | 0.523 | 0.861 | 0.717 | 0.154 | 0.198 | 0.797 | 0.725 |
| SeqDiffuSeq | 50M | 0.616 | 0.649 | 0.923 | 0.947 | 0.432 | 0.503 | 0.857 | 0.759 | 0.154 | 0.396 | 0.835 | 0.791 |
| *# Pretrain* | | | | | | | | | | | | | |
| DLM4G-1.o | 50M | 0.619 | 0.688 | 0.914 | 0.957 | 0.401 | 0.527 | 0.837 | 0.822 | 0.247 | 0.375 | 0.823 | 0.811 |
| DLM4G-2.o | 63M | **0.654** | **0.791** | **0.963** | **0.981** | **0.469** | **0.574** | **0.899** | **0.892** | **0.253** | **0.414** | **0.847** | **0.820** |
| %Gain | x1.5↑ | +4.1% | +14.9% | +4.3% | +2.5% | +8.5% | +8.9% | +4.4% | +8.5% | +2.4% | +10.4% | +2.9% | +1.1% |

## 4.3 FACTUAL GROUNDING AND EDIT SENSITIVITY

While the results on established metrics in Section 4.2 demonstrate our model's fluency, these scores are often insufficient for capturing the critical demands of G2S tasks: factual grounding to the source graph and sensitivity to its edits. To address this evaluation gap, we now introduce two novel, task-grounded metrics. To ensure a fair and direct comparison against the baseline results, we conduct this analysis on the WikiOFGraph dataset.

**Setup and Notations**: For the input KG ($\tilde{\mathcal{G}}$), we extract distinct entities as $\mathcal{U}_{\tilde{\mathcal{G}}} = \{h_i, t_j \mid (h_i, r_{ij}, t_j) \in \tilde{\mathcal{G}}\}$. For the corresponding generated sequence $\mathbf{S}$, we represent the extracted entities as $\mathcal{U}_{\mathbf{S}} = \{u \mid u \in \mathbf{S}\}$. Additionally, we maintain a hallucination set for the output: entities in $\mathbf{S}$ that are not members of $\mathcal{U}_{\tilde{\mathcal{G}}}$ constitute $\mathcal{H}_{\mathbf{S}}$ (with sequence length $N = |\mathbf{S}|$). For the entity and relation extraction, we use the alignment module discussed previously in section 3.3.

**Factual Grounding Metric (FGT, ↑)**: FGT measures how precisely the output realizes graph entities, with an optional penalty for out-of-graph mentions. We define Factual Grounding Metric (FGT) as:

$$\mathcal{F}_{\text{GT}}(\tilde{\mathcal{G}}, \mathbf{S}) = \underbrace{\frac{2\,|\mathcal{U}_{\tilde{\mathcal{G}}} \cap \mathcal{U}_{\mathbf{S}}|}{|\mathcal{U}_{\tilde{\mathcal{G}}}| + |\mathcal{U}_{\mathbf{S}}|}}_{\text{F1}} \left(1 - \lambda \frac{|\mathcal{H}_{\mathbf{S}}|}{N}\right). \tag{8}$$

We report results for $\lambda \in \{0, 0.5, 1\}$ and use $\lambda = 0.5$ by default, to balance the penalty term.

**Edit Sensitivity Rate (ESR, ↑)**: ESR is a precision focused metric. It evaluates whether the edits

in graph are realized in its generated sequence. Consider an original pair $(\tilde{\mathcal{G}}, \mathbf{S})$ and an edited pair $(\tilde{\mathcal{G}}', \mathbf{S}')$. We build $\mathcal{U}_{\tilde{\mathcal{G}}}, \mathcal{U}_{\tilde{\mathcal{G}}'}, \mathcal{U}_{\mathbf{S}}, \mathcal{U}_{\mathbf{S}'}$ as we do in FGT. The graph and text edits (e.g., additions or deletions) are defined as: $\Delta\mathcal{G} = (\mathcal{U}_{\tilde{\mathcal{G}}'} \setminus \mathcal{U}_{\tilde{\mathcal{G}}}) \cup (\mathcal{U}_{\tilde{\mathcal{G}}} \setminus \mathcal{U}_{\tilde{\mathcal{G}}'})$ and $\Delta\mathcal{T} = (\mathcal{U}_{\mathbf{S}'} \setminus \mathcal{U}_{\mathbf{S}}) \cup (\mathcal{U}_{\mathbf{S}} \setminus \mathcal{U}_{\mathbf{S}'})$. We define Edit Sensitivity Rate (ESR) as:

$$\mathcal{E}_{\mathrm{SR}}(\tilde{\mathcal{G}}, \mathbf{S}) = \frac{|\Delta\mathcal{G} \cap \Delta\mathcal{T}|}{|\Delta\mathcal{T}|}, \tag{9}$$

If the text does not change ($|\Delta\mathcal{T}| = 0$), set ESR $= 1$ when the graph also does not change ($|\Delta\mathcal{G}| = 0$) and ESR $= 0$ when the graph does change ($|\Delta\mathcal{G}| > 0$).

To evaluate FGT and ESR, we create edited graphs by randomly substituting a single entity with a plausible alternative from the vocabulary. We then measure whether the output text accurately reflects this specific modification. We compare `DLM4G` with comparably-size G2S models finetuned on the same task, and report FGT@{0, 0.5, 1} and ESR.

Table 5: Performance of `DLM4G` on Factual Grounding (FGT) and Edit Sensitivity (ESR).

| Model | Recall | F1 | $|\mathcal{H}_\mathbf{S}|$ | FGT@$\lambda$=0 | FGT@$\lambda$=0.5 | FGT@$\lambda$=1.0 | ESR |
|---|---|---|---|---|---|---|---|
| *# PLM baselines* | | | | | | | |
| GPT-2 (B) | 0.60 | 0.65 | 2.95 | 0.65 | 0.59 | 0.53 | 0.46 |
| T5 (S) | 0.58 | 0.62 | 3.10 | 0.62 | 0.56 | 0.50 | 0.42 |
| T5 (L) | 0.81 | 0.83 | 1.54 | 0.83 | 0.79 | 0.75 | 0.63 |
| `DLM4G`-1.0 | 0.80 | 0.79 | 2.03 | 0.79 | 0.74 | 0.70 | 0.60 |
| `DLM4G`-2.0 | **0.82** | **0.86** | **1.08** | **0.86** | **0.83** | **0.80** | **0.68** |
| % Gain (vs. T5-L) | +1.23% | +3.61% | 29.8% | +3.61% | +5.16% | +5.33% | +7.9% |

**Primary Findings**: Table 5, micro-averaged across 100 edited examples, highlights two key trends. *First*, among the baselines, T5-Large is the strongest, achieving the lowest hallucination rate (1.54 entities/sequence) and the best overall scores (FGT@0 of 0.83, FGT@0.5 of 0.79 and ESR of 0.63). *Second*, `DLM4G`-2.o consistently outperforms all baselines, improving upon T5-Large's recall (0.82 vs. 0.81) while reducing hallucinations by nearly 30% to a new low of 1.08 entities per sequence. Consequently, it achieves significant gains on our proposed metrics, improving the FGT score by +4.7% and the ESR score by +7.9%.

## 4.4 ABLATION

In this section we perform an ablation study to verify the impact of our graph-aware noise scheduling strategy and the choice of mapping function.

**Graph- aware schedule:** Table 6 compares the standard *sqrt* schedule against our proposed Graph-aware schedule. First, we observe that applying the Graph-aware schedule to *all* tokens yields an improvement (+0.03 BLEU). When we apply the Graph-aware schedule selectively to the graph-aligned tokens ($\mathcal{A}$) while keeping the standard schedule for others, performance improves significantly to **0.65 BLEU**. This suggests that the benefit comes not just from the schedule itself, but also from differentiating the noise profile of factual entities with syntactic text.

Table 6: Ablation of `DLM4G` noise schedules and mapping function $\Psi_i(x)$.

| DLM4G | Tokens | $\Psi_i(x)$ | B |
|---|---|---|---|
| *sqrt* (baseline) | All | linear | 0.60 |
| Graph-aware | All | linear | 0.63 |
| Graph-aware | $\mathcal{A}$ | linear | **0.65** |
| Graph-aware | $\mathcal{A}$ | poly | 0.61 |
| Graph-aware | $\mathcal{A}$ | cosine | 0.62 |

**Mapping Function**: We select the linear mapping primarily for its simplicity and ease of implementation. The impact of this design is visualized in Fig 3. As shown in Figure 3**L**, our adaptive schedule assigns a token-level noise schedule $\bar{\alpha}_t^i$ compared to the global *sqrt* baseline. As seen by contrasting the loss profiles of syntactic tokens (Fig 3**M**) and factual tokens (Fig 3**R**), factual entities exhibit higher reconstruction difficulty but, under our graph-aware schedule, their loss evolves in a smooth, approximately linear over time. This stabilized, monotone difficulty profile keeps factual tokens informative throughout the trajectory and allows the denoiser to recover them more faithfully. While we also explored exponential, cosine, and polynomial mappings (Table 6), we found they offered no performance benefit. Detailed comparisons of these alternatives are in App. A.7.

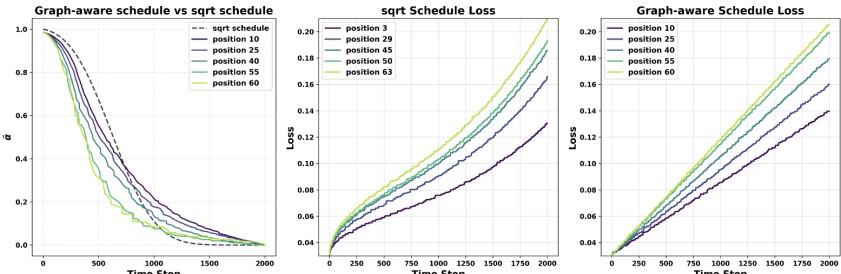

Figure 3: (**Left**) Noise schedule corresponding to alignment set $i \in \mathcal{A}$ (position 10, 25,.., 60) compared against *sqrt* schedule for $i \notin \mathcal{A}$ (position 3, 29,.., 63); (**Mid**) The loss profile across time steps ($t = 0 \rightarrow T$) for the syntactic tokens (**Right**) The loss profile across time steps ($t = 0 \rightarrow T$) for the factual tokens.

## 4.5 `DLM4G` FOR MOLECULE CAPTIONING

`DLM4G` has demonstrated strong performance in fluency (Section 4.2) and factual grounding (Section 4.3). We now test its generalization to a complex, real-world application by applying it to molecule captioning—a challenging Graph-to-Sequence task from the scientific domain. This benchmark evaluates whether our model's efficient, graph-aware design can outperform larger, specialized models in a completely different field, demonstrating its practical utility

**Dataset and Graph representation**: We use a subset of the M3-20M dataset Guo et al. (2025) containing 360,000 SMILES-description pairs, which we split 80/10/10 for training, validation, and testing. To process this data, we convert each SMILES string into a knowledge graph $\tilde{\mathcal{G}}$, where the molecule's atoms are treated as entities (nodes) and the chemical bonds between them are the relations (edges). This allows our model to directly interpret the molecule's topology.

**Results**: First we analyze the scaling effect within the `DLM4G` variants. As shown in Fig 7, the larger `DLM4G`-2.o (63M parameters) consistently outperforms the `DLM4G`-1.o version (50M). It achieves a +6.1% improvement in BLEU, a +2.6% gain in chrF++, and a significant +11.7% increase in METEOR. This validates our scaling approach and establishes `DLM4G`-2.o as our best model.

More importantly, `DLM4G`-2.o achieves a new state-of-the-art result against all specialized baselines. The detailed analysis beside the table 7 highlights the specific performance gains and the model's remarkable parameter efficiency. Refer Appendix A.9 for more details.

Table 7: Comparison of our `DLM4G` models against baselines.

| Method | #P | B | CrF++ | M | B-F1 | MVE |
|---|---|---|---|---|---|---|
| MolT5 (B) | 220M | 0.452 | 0.651 | 0.510 | 0.681 | 0.852 |
| GitMol | 700M | 0.475 | 0.680 | 0.532 | 0.751 | 0.875 |
| GraphT5 | 272M | 0.481 | 0.692 | 0.545 | 0.810 | 0.913 |
| DLM4G-1.o | 50M | 0.534 | 0.715 | 0.560 | 0.816 | 0.901 |
| DLM4G-2.o | 63M | **0.567** | **0.734** | **0.626** | **0.843** | **0.925** |
| %Gain | x12↑ | +17.8% | +6.1% | +14.8% | +4.1% | +1.3% |

**Analysis of Results:** Our `DLM4G`-2.o model outperforms all baselines across every metric. It demonstrates strong performance on surface-level scores, achieving a BLEU of 0.567 (a +17.8% gain over the best baseline), and also leads on semantic metrics with a BERTScore-F1 of 0.843. Crucially, it delivers these results while being 4x to 11x smaller than the baselines.

## 5 CONCLUSION AND LIMITATIONS

We presented `DLM4G`, a graph-conditioned, non-autoregressive diffusion framework for graph-to-sequence generation that targets two persistent failures of PLMs—factual grounding and edit sensitivity. Our approach learns a graph-aware noising schedule that prioritizes graph-aligned tokens during training, and at inference combines this schedule with cross-attention to the graph to guide denoising. Across standard surface and embedding metrics, `DLM4G` surpasses strong baselines; on two task-grounded metrics, it outperforms comparably sized models. Extending to molecule captioning further demonstrates generality. While promising, `DLM4G` introduces diffusion-time costs and relies on entity alignment quality and operates on a fixed, dataset-defined linearization of the input KG. As a result, the model is not permutation invariant and may in principle be sensitive to alternative serialization schemes. Future work will reduce sampling steps, relax alignment dependence, and explore structure-aware encoding.

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

# A APPENDIX

This section presents an in-depth discussion of the eleven core components of the manuscript, including the principal mathematical derivations, template methods (zero-shot prompting and molecular captioning), the proposed algorithm pseudo-codes, and detailed implementation aspects. Additionally, the complete code implementation is available here: `CODE`

## A.1 DERIVATION OF THE TRAINING OBJECTIVE

`DLM4G` builds on the standard diffusion framework, which trades the flexibility of expressive generative models (e.g., GANs, VAEs, flow models) for the tractability of likelihood-based training in a continuous latent space $\mathbf{z}$. The overall goal is to minimize the negative log-likelihood

$$\mathbb{E}_{\mathbf{z}_0, \mathbf{c}}\big[-\log p_\theta(\mathbf{z}_0 \mid \mathbf{c})\big], \tag{10}$$

which is upper-bounded by the Variational Lower Bound (VLB).

### A.1.1 FORWARD AND REVERSE PROCESSES

The forward Markov chain is defined as $q(\mathbf{z}_{1:T} \mid \mathbf{z}_0) = \prod_{t=1}^{T} q(\mathbf{z}_t \mid \mathbf{z}_{t-1})$, where each transition is Gaussian:

$$q(\mathbf{z}_t \mid \mathbf{z}_{t-1}) = \mathcal{N}\Big(\mathbf{z}_t \mid \sqrt{1 - \beta_t}\, \mathbf{z}_{t-1},\, \beta_t\, \mathbf{I}\Big). \tag{11}$$

Let $\alpha_t = 1 - \beta_t$ and $\bar{\alpha}_t = \prod_{i=1}^{t} \alpha_i$. By induction, the marginal at time $t$ satisfies:

$$\mathbf{z}_t = \sqrt{\bar{\alpha}_t}\, \mathbf{z}_0 + \sqrt{1 - \bar{\alpha}_t}\, \boldsymbol{\epsilon}, \qquad \boldsymbol{\epsilon} \sim \mathcal{N}(\mathbf{0}, \mathbf{I}), \tag{12}$$

so that $q(\mathbf{z}_t \mid \mathbf{z}_0) = \mathcal{N}\Big(\sqrt{\bar{\alpha}_t}\, \mathbf{z}_0,\, (1 - \bar{\alpha}_t)\mathbf{I}\Big)$. We used the *sqrt* schedule as the baseline schedule used in DiffusionLM Li et al. (2022), namely $\bar{\alpha}_t = 1 - \sqrt{t/T + s}$ with small $s > 0$. The reverse denoising process then learns

$$p_\theta(\mathbf{z}_{0:T}) = p(\mathbf{z}_T) \prod_{t=1}^{T} p_\theta(\mathbf{z}_{t-1} \mid \mathbf{z}_t), \quad p_\theta(\mathbf{z}_{t-1} \mid \mathbf{z}_t) = \mathcal{N}\big(\boldsymbol{\mu}_\theta(\mathbf{z}_t, t),\, \boldsymbol{\sigma}_\theta^2(\mathbf{z}_t, t)\big). \tag{13}$$

Applying Bayes' rule to the forward transitions yields the exact posterior mean

$$\boldsymbol{\mu}_t(\mathbf{z}_t, \mathbf{z}_0) = \frac{\sqrt{\alpha_t}(1 - \bar{\alpha}_{t-1})}{1 - \bar{\alpha}_t}\, \mathbf{z}_t + \frac{\sqrt{\bar{\alpha}_{t-1}}\, \beta_t}{1 - \bar{\alpha}_t}\, \mathbf{z}_0, \tag{14}$$

whose coefficients we denote by $\mathcal{U}$ and $\mathcal{E}$. `DLM4G`'s training objective is then to match the network's predicted $\boldsymbol{\mu}_\theta, \boldsymbol{\sigma}_\theta$ to these posterior quantities via a simple noise-prediction loss. We optimize the negative log-likelihood by upper-bounding it with the variational lower bound

$$\mathbb{E}\big[-\log p_\theta(x_0)\big] \leq \mathcal{L}_{\text{vlb}} = \sum_{t=0}^{T} \mathcal{L}_t. \tag{15}$$

### A.1.2 VARIATIONAL LOWER BOUND (VLB)

Following Sohl-Dickstein et al.Sohl-Dickstein et al. (2015), for conditional generation the VLB decomposes into:

$$\mathcal{L}_{\text{vlb}} = \mathbb{E}_{q(\mathbf{z}_{1:T}|\mathbf{z}_0)}\left[\underbrace{\log \frac{q(\mathbf{z}_T \mid \mathbf{z}_0)}{p(\mathbf{z}_T)}}_{\mathcal{L}_T} + \sum_{t=2}^{T} \underbrace{\log \frac{q(\mathbf{z}_{t-1} \mid \mathbf{z}_t, \mathbf{z}_0)}{p_\theta(\mathbf{z}_{t-1} \mid \mathbf{z}_t, \mathbf{c})}}_{\mathcal{L}_t} - \underbrace{\log p_\theta(\mathbf{z}_0 \mid \mathbf{z}_1, \mathbf{c})}_{\mathcal{L}_0}\right], \tag{16}$$

where each $\mathcal{L}_t$ is a KL divergence between Gaussians. The true posterior mean (via Bayes' rule) is:

$$\boldsymbol{\mu}_t(\mathbf{z}_t, \mathbf{z}_0) = \underbrace{\frac{\sqrt{\alpha_t}(1 - \bar{\alpha}_{t-1})}{1 - \bar{\alpha}_t}}_{\mathcal{U}}\, \mathbf{z}_t + \underbrace{\frac{\sqrt{\bar{\alpha}_{t-1}}\beta_t}{1 - \bar{\alpha}_t}}_{\mathcal{E}}\, \mathbf{z}_0, \tag{17}$$

with covariance $\boldsymbol{\Sigma}_q = \tilde{\beta}_t \mathbf{I}$, $\tilde{\beta}_t = \frac{1-\bar{\alpha}_{t-1}}{1-\bar{\alpha}_t}\beta_t$. In the standard simplification, the model's covariance is fixed to match the true posterior covariance $\boldsymbol{\Sigma}_\theta = \boldsymbol{\Sigma}_q$, the KL collapses to a weighted MSE:

$$\mathcal{L}_t = \frac{1}{2}\big\|\boldsymbol{\mu}_t - \boldsymbol{\mu}_\theta\big\|^2_{\boldsymbol{\Sigma}_q^{-1}} \; \propto \; \mathbb{E}\Big[\big\|\mathbf{z}_0 - \mathcal{M}_\theta(\mathbf{z}_t, t, \mathbf{c})\big\|^2\Big]. \tag{18}$$

Thus, for $2 \le t \le T$, $\mathcal{L}_t \to \|\mathbf{z}_0 - \mathcal{M}_\theta(\mathbf{z}_t, t, \mathbf{c})\|^2$. The final KL encourages $\mathbf{z}_T$ to match the unit Gaussian prior:

$$\mathcal{L}_T = \mathrm{KL}\big(q(\mathbf{z}_T \mid \mathbf{z}_0)\,\|\,p(\mathbf{z}_T)\big) \; \propto \; \big\|\boldsymbol{\mu}(\mathbf{z}_T)\big\|^2, \tag{19}$$

a constant w.r.t. $\theta$. The discrete target $\mathbf{S}$ (sequence) is encoded into a continuous embedding $g_\Phi(\mathbf{S})$. The final term in VLB is $\mathcal{L}_0 = -\log p_\theta(\mathbf{z}_0 \mid \mathbf{z}_1, \mathbf{c})$. We need to integrate the discrete data $\mathbf{S}$ into this continuous likelihood term. We use the law of total probability to express the continuous likelihood $p_\theta(\mathbf{z}_0 \mid \mathbf{z}_1, \mathbf{c})$ by marginalizing over all possible discrete tokens in the target sequence $\mathbf{S} = \{s_1, s_2 \cdots, s_N\}$:

$$p_\theta(\mathbf{z}_0 \mid \mathbf{z}_1, \mathbf{c}) = \sum_{\mathbf{S}} p_\theta(\mathbf{z}_0, \mathbf{S} \mid \mathbf{z}_1, \mathbf{c}) \tag{20}$$

We then apply the product rule to the joint probability:

$$p_\theta(\mathbf{z}_0, \mathbf{S} \mid \mathbf{z}_1, \mathbf{c}) = p_\theta(\mathbf{z}_0 \mid \mathbf{S}, \mathbf{z}_1, \mathbf{c}) \cdot p_\theta(\mathbf{S} \mid \mathbf{z}_1, \mathbf{c}) \tag{21}$$

For training, we are interested in the specific ground-truth sequence $\mathbf{S}$. When we evaluate $\mathcal{L}_0$ during training, we consider only the term where $\mathbf{S}$ is the ground-truth sequence:

$$\mathcal{L}_0 \approx -\log p_\theta(\mathbf{z}_0, \mathbf{S} \mid \mathbf{z}_1, \mathbf{c}) = -\log\left[p_\theta(\mathbf{z}_0 \mid \mathbf{S}, \mathbf{z}_1, \mathbf{c}) \cdot p_\theta(\mathbf{S} \mid \mathbf{z}_1, \mathbf{c})\right] \tag{22}$$

The core approximation simplifies the dependency graph by asserting that the discrete data $\mathbf{S}$ is generated only from the clean latent $\mathbf{z}_0$, and is independent of $\mathbf{z}_1$ and $\mathbf{c}$ given $\mathbf{z}_0$.

$$\mathbf{S} \perp (\mathbf{z}_1, \mathbf{c}) \mid \mathbf{z}_0 \tag{23}$$

This allows us to replace the discrete conditional likelihood with the separate rounding network $\tilde{p}_\Phi(\mathbf{S} \mid \mathbf{z}_0)$: $p_\theta(\mathbf{S} \mid \mathbf{z}_1, \mathbf{c}) \approx \tilde{p}_\Phi(\mathbf{S} \mid \mathbf{z}_0)$. Substituting this back into the likelihood decomposition:

$$p_\theta(\mathbf{z}_0, \mathbf{S} \mid \mathbf{z}_1, \mathbf{c}) \approx p_{\mathrm{cont}}(\mathbf{z}_0 \mid \mathbf{S}, \mathbf{z}_1, \mathbf{c}) \cdot \tilde{p}_\Phi(\mathbf{S} \mid \mathbf{z}_0) \tag{24}$$

Taking the negative logarithm of the approximation gives the two desired terms:

$$\mathcal{L}_0 \approx -\log p_{\mathrm{cont}}(\mathbf{z}_0 \mid \mathbf{S}, \mathbf{z}_1, \mathbf{c}) - \log \tilde{p}_\Phi(\mathbf{S} \mid \mathbf{z}_0)$$

This split yields the two components used in the final training objective:

1. Consistency Term ($\mathcal{L}_{\mathrm{Cons}}$): The first term is the negative log-likelihood of the continuous latent, which is minimized via the MSE loss on the means: $-\log p_{\mathrm{cont}}(\mathbf{z}_0 \mid \mathbf{S}, \mathbf{z}_1, \mathbf{c}) \to$ $\mathcal{L}_{\mathrm{Consistency}} = \big\|g_\Phi(\mathbf{S}) - \mathcal{M}_\theta(\mathbf{z}_1, 1, \mathbf{c})\big\|^2$.
2. Rounding Term ($\mathcal{L}_{\mathrm{Round}}$): This second term is the dedicated loss for the discrete data likelihood: $\mathcal{L}_{\mathrm{Round}} = -\log \tilde{p}_\Phi(\mathbf{S} \mid \mathbf{z}_0)$

### A.1.3 FINAL END-TO-END OBJECTIVE

Combining all components:

$$\mathcal{L}_{\mathrm{vlb}} \propto \sum_{t=2}^{T} \underbrace{\big\|\mathbf{z}_0 - \mathcal{M}_\theta(\mathbf{z}_t, t, \mathbf{c})\big\|^2}_{\text{Denoising}} + \underbrace{\big\|g_\Phi(\mathbf{S}) - \mathcal{M}_\theta(\mathbf{z}_1, 1, \mathbf{c})\big\|^2}_{\text{Consistency}} - \underbrace{\log \tilde{p}_\Phi(\mathbf{S} \mid \mathbf{z}_0)}_{\text{Rounding}}. \tag{25}$$

Dropping constant terms, the simplified end-to-end training loss is:

$$\mathcal{L}_{\mathrm{e2e\text{-}simple}}(\mathbf{S}) = \mathbb{E}_q\Big[\sum_{t=2}^{T} \underbrace{\|\mathcal{M}_\theta(\mathbf{z}_t, t, \tilde{\mathcal{G}}) - \mathbf{z}_0\|^2}_{\text{Denoising}} + \underbrace{\|g_\Phi(\mathbf{S}) - \mathcal{M}_\theta(\mathbf{z}_1, 1, \tilde{\mathcal{G}})\|^2}_{\text{Consistency}} - \underbrace{\log \tilde{p}_\Phi(\mathbf{S} \mid \mathbf{z}_0)}_{\text{Rounding}}\Big] \tag{26}$$

## A.2 RELATED WORK AND BACKGROUND

**Graph-to-Sequence Learning**: *G2S* has evolved through three stages: (i) template-based systems that verbalised graph predicates but were brittle for complex inputs Wiseman et al. (2018); Kasner & Dusek (2022); Vejvar & Fujimoto (2023); (ii) neural encoder–decoder models that learned graph embeddings, improving structural generalisation yet struggling with long-range dependencies Wiseman et al. (2017); Beck et al. (2018); Iso et al. (2019); and (iii) fine-tuned transformers, now dominant, offering superior fluency and factuality with minimal task-specific design Vaswani et al. (2023); Ribeiro et al. (2021); Jolly et al. (2021); Han & Shareghi (2022). This trajectory frames the current *G2S* landscape and motivates subsequent approaches.

**PLMs for Graphs**: Leveraging LLMs for graph verbalisation involves four challenges: (i) *alignment* of graph elements to words Luo et al. (2024); Zhu et al. (2025), (ii) *position* encoding under permutation invariance Black et al. (2024); Huang et al. (2024); Perozzi et al. (2024), (iii) *multi-level semantics* across nodes, edges, and subgraphs Wang et al. (2024a), and (iv) *context* retention over long spans Ding et al. (2025); Wang et al. (2024c). These define a taxonomy from Graph-to-Sequence (G2S) to Graph-to-Token (G2T) methods. Current KG-to-text models employ positional encodings, structural prompts, and multi-granularity attention Luo et al. (2024); Zhu et al. (2025); Wang et al. (2024a), reducing factual omissions but still limited by left-to-right decoding and weak global planning Wei et al. (2022); Lin et al. (2021). Diffusion LMs, with iterative denoising, could address these issues, though they remain unexplored for KG-to-text generation Li et al. (2023).

**Diffusion Models for Conditional Generation**: Conditional diffusion guides denoising with an input sequence encoding, extending conditional-VAE ideas Zhao et al. (2017). Early text models (Diffusion-LM Li et al. (2022), Analog Bits Chen et al. (2023)) imposed weak conditioning via classifiers or plug-in controls, while DIFFUSEQ Gong et al. (2023); Yuan et al. (2024) enabled true sequence-to-sequence conditioning in continuous space. Related frameworks also target time-series (CSDI Tashiro et al. (2021)) and speech (WaveGrad Chen et al. (2021b)). Distinct from prior *G2S* and diffusion-LM work, `DLM4G` integrates classifier-free diffusion with explicit KG conditioning, treating the graph itself as the control variable. This eliminates exposure bias and supports global planning, yielding more coherent KG verbalisation.

**Molecule Captioning:** Most prior works adapt either AR or NAR generation for molecular descriptions, but these methods often inherit exposure bias (AR) or strong independence assumptions (NAR) Edwards et al. (2022); Liu et al. (2024a). Diffusion-based approaches, while promising for text generation, have not been systematically applied to graph-to-sequence captioning. To clarify the conceptual distinctions, Table 8 summarizes the characteristics of major generation paradigms and highlights how `DLM4G` differs. In particular, our method introduces a *graph-guided refinement process* with *graph-aware noising*, enabling both factual grounding and graph edits during caption generation, a capability absent in existing paradigms.

Table 8: Comparison of `DLM4G` with existing paradigms (FG: Factual Grounding; GE: Graph Edits).

| Model Family | Output Generation Paradigm | Noising Schedule | Mechanism for FG / GE | Molecule Captioning |
|---|---|---|---|---|
| **Autoregressive (AR)** | Sequential, left-to-right token prediction *(Exposure bias, local optima, e.g., BART, T5)* | No Diffusion | Implicit (data-driven/ graph prompts ) | Standard G2S application |
| **Non-Autoregressive (NAR)** | Parallel, independent token prediction *(Conditional independence assumption) (e.g., Mask-Predict)* | No Diffusion | Implicit (sequence-level objective; no graph-aware bias) | Standard G2S application |
| **Standard Diffusion LMs** | Iterative, parallel refinement from noise *Advantage: Mitigates exposure bias(e.g., DiffuSeq)* | Uniform, Isotropic | Implicit (standard diffusion schedule) | Unexplored for G2S; applied to S2G (generation) |
| **DLM4G (Ours)** | **Iterative, graph-guided refinement** *(Global planning + factual grounding)* | **Graph-aware noising** *(Preserves entity, relations)* | **Explicit (graph-aware schedule over $\mathcal{A}$)** | **Novel G2S application** *(Graph → Sequence task)* |

## A.3 SUMMARY OF DATASET AND BASELINES

*WikiOFGraph*: We use the WikiOFGraph dataset as described in Kim et al. (2024). This dataset comprises approximately 5.85 million graph–text pairs extracted from general-domain English Wikipedia articles. Each graph is represented as a set of RDF-style triples, automatically mined and refined via large-language-model prompting. For example, the triple <Alan Turing, birthPlace, London> corresponds to the sentence "Alan Turing was born in London."

Table 9: Training set statistics for comparative analysis. *# triplet (m/M/avg)* indicates the minimum, maximum, and average number of triplets per sample.

| Dataset | # samples | # unique predicate | # unique entity | # triplet (m/M/avg) |
|---|---|---|---|---|
| WikiOFGraph | 5.85M | 140,733 | 8.2M | 1/173/3.62 |
| GenWiki | 680K | 287 | 86.6K | 1/10/2.64 |
| TekGen | 6.31M | 50,861 | 4.3M | 1/54/1.73 |

*GenWiki*: We use the "fine" split of GenWiki Jin et al. (2020), which contains 680 K graph–text pairs; we reserve 10 % of these for evaluation. The dataset covers 287 distinct predicates, with an average of $2.64 \pm 1.72$ triples per graph and an average text length of $26.05 \pm 10.99$ tokens. For instance, the graph { (Google, founder, Larry Page), (Google, founder, Sergey Brin)} maps to the sentence "Google was founded by Larry Page and Sergey Brin."

*TekGen*: We adopt the TekGen dataset as released in Mousavi et al. (2024), containing roughly 6.3 M aligned Wikidata triple–sentence pairs drawn from Wikipedia. It spans about 50.8 K distinct predicates and is provided in separate train/validation/test TSV files (each line a JSON object). An exemplar entry is: *{"subject":"The Lion King","predicate":"director","object":"Roger Allers","text":"The Lion King is an animated musical drama film directed by Roger Allers and Rob Minkoff."}*

### A.4 ALIGNMENT MODULE

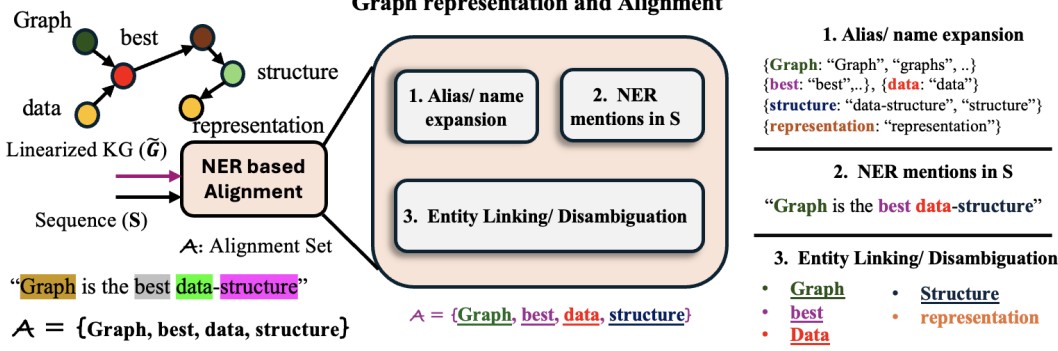

Figure 4: NER-based graph–sequence alignment. Given a linearized graph $\tilde{\mathcal{G}}$ and target sequence $\mathbf{S}$, the module (1) expands aliases for each KG node, (2) detects mentions in $\mathbf{S}$, and (3) links/disambiguates them to obtain the alignment set $\mathcal{A}$ of graph-grounded tokens.

**Setup.** For each graph–sequence pair $(\tilde{\mathcal{G}}, \mathbf{S})$, the alignment module outputs a set $\mathcal{A}$ of token spans in $\mathbf{S}$ that are linked to entities or relations in $\tilde{\mathcal{G}}$ (see Fig. 4 and Sec. 3.3). To quantify the quality of this mapping, we manually annotate a subset of WikiOFGraph dev examples (100 examples) with gold alignments $\mathcal{A}^\star$ and evaluate the module at the span level: a prediction is correct (TP) if the span overlaps a gold mention and is linked to the same KG node; other predicted spans are counted as FP, and unmatched gold spans as FN. We report precision, recall, and F1 over (span, KG node) pairs, as well as token and KG-node coverage.

**Overall quality.** Table 10 summarizes the quality of the alignment module for our default configuration (maximum of $k=5$ aliases per KG node). Specifically, **Token coverage** measures the percentage of tokens in the target sequence $\mathbf{S}$ that are part of an aligned span, while **KG-node coverage** measures the percentage of entities and relations in $\tilde{\mathcal{G}}$ that successfully link to $\mathbf{S}$. The module achieves high precision while covering a substantial fraction of graph-grounded tokens and KG nodes, which is sufficient to anchor the graph-aware noising schedule.

**Effect of alias-set size.** The size of the alias set controls the effective size of $\mathcal{A}$: larger $k$ exposes more surface forms and increases recall and coverage, but can introduce additional ambiguity and harm precision. We vary $k \in \{2, 3, 4, 5\}$ and re-evaluate the module on the same annotated subset, as

Table 10: Alignment quality on the WikiOFGraph (dev), with $k=5$ aliases per KG node. The average magnitude of the alignment set ($|\mathcal{A}|$) represents the size of alignment set per example. Token coverage is the percentage of target tokens that belong to some aligned span; KG-node coverage is the percentage of nodes in $\tilde{\mathcal{G}}$ with at least one aligned mention in $\mathbf{S}$.

| Setting | Prec. | Rec. | F1 | Token cov. (%) | KG-node cov. (%) | $|\mathcal{A}|$ |
|---------|-------|------|------|----------------|------------------|-----------------|
| Aliases ($k=5$) | 0.90 | 0.78 | 0.83 | 24.1 | 86.3 | 7.4 |

well as the downstream performance of `DLM4G` on WikiOFGraph (Table 12). We observe a smooth precision–recall trade-off as $k$ increases; the default $k=5$ offers a good balance, yielding the best BLEU score.

Table 11: Alias-budget ablation on WikiOFGraph. Increasing the maximum number of aliases $k$ per KG node improves recall and coverage but slightly reduces precision, resulting in a modest but consistent gain in downstream performance.

| $k$ (aliases) | Prec. | Rec. | F1 | $|\mathcal{A}|$ | Token cov. (%) | KG-node cov. (%) | BLEU |
|---------------|-------|------|------|-----------------|----------------|------------------|-------|
| 2 | 0.94 | 0.70 | 0.80 | 3.9 | 17.3 | 75.1 | 0.603 |
| 3 | 0.92 | 0.75 | 0.83 | 4.3 | 20.5 | 80.4 | 0.609 |
| 4 | 0.91 | 0.77 | 0.84 | 6.7 | 22.8 | 84.0 | 0.624 |
| 5 | 0.90 | 0.78 | 0.83 | 7.4 | 24.1 | 86.3 | 0.651 |

**Example.** To illustrate the alignment process, consider the following graph–sequence pair:
Serialized KG ($\tilde{\mathcal{G}}$): $\langle$`[HEAD] USA [REL] hosted [TAIL] 1994_FIFA_World_Cup`$\rangle$ `[SEP]` $\langle$`[HEAD] USA [REL] capital [TAIL] Washington_D.C.`$\rangle$ `[SEP]` $\langle$`[HEAD] 1994_FIFA_World_Cup [REL] top_scorer [TAIL] Hristo_Stoichkov`$\rangle$.
Corresponding sequence ($\mathbf{S}$): *"The United States hosted the 1994 FIFA World Cup; its capital is Washington, D.C., and the tournament's top scorer was Hristo Stoichkov"*.
*(1) Alias expansion.* From the KG we construct an alias dictionary, e.g.,

$$\text{USA: } \{\text{"USA", "U.S.", "United States", "United States of America", \dots}\},$$
$$\text{1994\_FIFA\_World\_Cup: } \{\text{"1994 FIFA World Cup", "1994 World Cup", \dots}\},$$
$$\text{Washington\_D.C.: } \{\text{"Washington, D.C.", "Washington DC", \dots}\},$$
$$\text{Hristo\_Stoichkov: } \{\text{"Hristo Stoichkov", "Stoichkov"}\}.$$

*(2) NER mentions in $\mathbf{S}$.* A NER detector identifies mentions such as "United States", "1994 FIFA World Cup", "Washington, D.C.", and "Hristo Stoichkov" in the sequence.
*(3) Entity linking / disambiguation.* Each mention is matched against the alias dictionary and, if multiple candidates exist, disambiguated using local context similarity to KG node descriptions. For this example the module recovers the alignment set $\mathcal{A}$ as:
$\mathcal{A} = \{$("United States", USA), ("1994 FIFA World Cup", 1994_FIFA_World_Cup), ("Washington, D.C.", Washington_D.C.), ("Hristo Stoichkov", Hristo_Stoichkov), ("hosted", hosted), ("capital", capital), ("top scorer", top_scorer)$\}\}$,
which corresponds to seven true-positive links between $\mathbf{S}$ and $\tilde{\mathcal{G}}$. This alignment set is then used to derive token-wise difficulty profiles and the graph-aware noising schedule described in Sec. 3.3.
**Details.** For this specific instance, the sequence $\mathbf{S}$ contains 32 tokens. The $\tilde{\mathcal{G}}$ contains 7 unique factual elements (4 entities and 3 relations). The alignment set $\mathcal{A}$ has a magnitude $|\mathcal{A}| = \mathbf{7.0}$, aligning all 7 elements (100.0% KG-node coverage). The results are reported in Table 12.

Table 12: Alignment Metrics for the example $(\tilde{\mathcal{G}}, \mathbf{S})$ pair .

| $k$ (aliases) | Prec. | Rec. | F1 | $|\mathcal{A}|$ | Token cov. (%) | KG-node cov. (%) | BLEU |
|---------------|-------|------|------|-----------------|----------------|------------------|-------|
| 5 | 1.00 | 1.00 | 1.00 | 7.0 | 31.2 | 100.0 | 0.773 |

### A.5 NON- INCREASING ISOTONIC PROJECTION

After constructing the per-token cumulative schedule $\tilde{\alpha}_t^i$, we project it onto the set of non-increasing sequences $\{\bar{\alpha}_t^i\}_{t=1}^T$ such that $\bar{\alpha}_1^i \geq \bar{\alpha}_2^i \geq \cdots \geq \bar{\alpha}_T^i$. Concretely, this is a 1D isotonic regression problem with squared loss, which we solve using the standard Pool-Adjacent-Violators Algorithm (PAVA). This algorithm finds the closest monotone non-increasing sequence (in the least-squares sense) to the input. Intuitively, it smooths out spurious "bumps" in the loss profile while guaranteeing that the cumulative signal strength strictly decays over time, fulfilling the monotonicity requirement of the diffusion process.

### A.6 TRAINING DETAILS

*Model variants*: We train two Transformer–based denoisers: (i) a 6-encoder / 6-decoder architecture with $\approx 50$ M parameters, and (ii) a 6-encoder / 9-decoder architecture with $\approx 63$ M parameters. Both use GeLU activations Vaswani et al. (2023); Hendrycks & Gimpel (2023) and share all other hyper-parameters.

*Diffusion setup*: A fixed diffusion horizon of $T = 2000$ timesteps is employed, following the *sqrt* noise schedule introduced in DiffusionLM Li et al. (2022). Inputs are tokenised with the `bert-base-uncased` vocabulary Devlin et al. (2019). The graph-aware noising schedule is calculated every 20,000 training steps.

*Optimization*: All experiments use AdamW with a peak learning rate of $1 \times 10^{-4}$, a linear warm-up of 10,000 steps, and linear decay to zero. Gradient norms are clipped to 1.0; no label-smoothing or dropout is applied beyond the architectural dropout already reported in the main text.

*Training regime*: Each model is trained for up to 200,000 steps per dataset:

- The 50 M model achieves its best validation metrics after $\sim 190,000$ steps.
- The 63 M model converges at the full 200,000-steps budget.

These numbers were found to be stable across all datasets considered.

### A.7 MAPPING FUNCTION ABLATIONS

**Choice of Mapping Function:** As discussed in Sec. 3.3, the graph-aware schedule is obtained by mapping token-wise difficulty profiles $\{\ell_t^i\}_{t=1}^T$ through a monotone mapping function $\Psi_i(x)$ to produce the cumulative noise schedule $\{\bar{\alpha}_t^i\}_{t=1}^T$. For $x \in [\ell_{t-1}^i, \ell_t^i]$ we write

$$\Psi_i(x) = \bar{\alpha}_{t-1} + (\bar{\alpha}_t - \bar{\alpha}_{t-1}) \, \phi\left(\frac{x - \ell_{t-1}^i}{\ell_t^i - \ell_{t-1}^i}\right), \qquad t = 2, \ldots, T, \tag{27}$$

In the main experiments we use a linear mapping $\phi_{\text{lin}}(u) = u$. Here we ablate three smooth alternatives—exponential, cosine, and polynomial mappings, all with the same boundary conditions. Table 14 summarizes the functional forms, and Fig. 5(**L–R**) visualizes the induced loss profiles over diffusion steps for aligned token positions (10, 25, 40, 55, 60). All three mappings introduce sharper non-linearities (e.g., flatter early regions and steeper tails), which make the token-wise loss grow more abruptly near the end of the diffusion process. Empirically, this concentration of noise updates degrades downstream performance: we observe small but consistent drops in **BLEU** (see Table 6). Hence, we retain the simple linear mapping $\phi_{\text{lin}}(u) = u$ in `DLM4G`.

**Impact on Performance:** In addition to BLEU, we assess how the choice of mapping function affects factual grounding and edit sensitivity on WikiOFGraph, using the metrics introduced in Sec. 4.3. For FGT we report FGT@$\lambda = 0.5$, our default setting which balances the penalty on hallucinated entities; ESR has no hyperparameter. Table 13 extends the mapping ablation to these metrics. We find that the graph-aware linear mapping over aligned tokens $\mathcal{A}$ improves BLEU and FGT while also achieving the highest ESR, whereas the more non-linear polynomial and cosine mappings consistently degrade all three metrics. This supports our qualitative analysis in Fig. 5 and further motivates our choice of the simple linear mapping in `DLM4G`.

Table 13: Ablation of `DLM4G` noise schedules and mapping function $\Psi_i(x)$ on WikiOFGraph. We report BLEU (**B**), Factual Grounding (FGT@$\lambda = 0.5$, higher is better), and Edit Sensitivity Rate (ESR, higher is better).

| `DLM4G` | Tokens | $\Psi_i(x)$ | B | FGT@0.5 | ESR |
|---|---|---|---|---|---|
| Graph-aware | $\mathcal{A}$ | linear | **0.65** | **0.83** | **0.68** |
| Graph-aware | $\mathcal{A}$ | poly | 0.61 | 0.80 | 0.61 |
| Graph-aware | $\mathcal{A}$ | cosine | 0.62 | 0.80 | 0.63 |

Table 14: Ablation of $\phi(u)$ used in the graph-aware schedule.

| Ablation | $\phi(u)$ |
|---|---|
| Linear | $\phi_{\text{lin}}(u) = u$ |
| Polynomial | $\phi_{\text{poly}}(u) = u^p \quad$ (we use $p = 2$) |
| Exponential | $\phi_{\text{exp}}(u) = \dfrac{e^{\beta u} - 1}{e^{\beta} - 1} \quad$ (we use $\beta = 3$) |
| Cosine | $\phi_{\text{cos}}(u) = \frac{1}{2}\big(1 - \cos(\pi u)\big)$ |

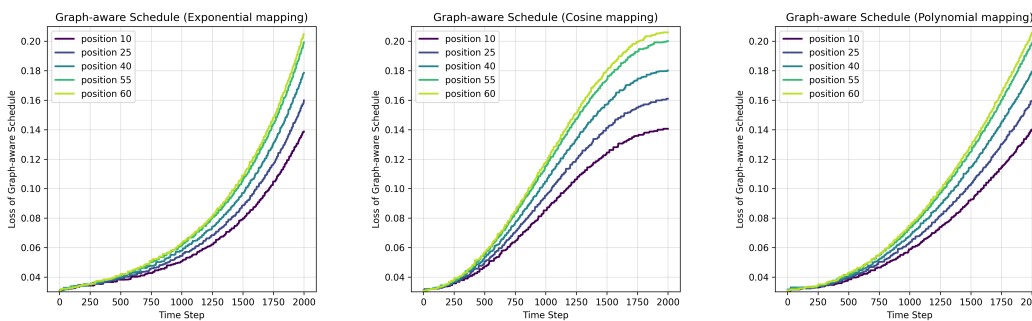

Figure 5: Induced loss profiles for the graph-aware schedule under different mappings: (**L**eft) exponential, (**M**id) cosine, and (**R**ight) polynomial. Curves show loss trajectories for aligned token positions at steps 10, 25, 40, 55, and 60.

```
==== System Prompt====                          ==== System Prompt====
You are {MODEL}, a large language model.        You are {GPT-o4-mini}, a large language model.
Your task is to convert a flat list of RDF-style triples into a single,   Your task is to convert a flat list of RDF-style triples into a single, fluent
fluent English description.                     English description.

==== MODEL-SPECIFIC GUIDANCE ====               ==== MODEL-SPECIFIC GUIDANCE ====
{MODEL_GUIDANCE}                                {• Keep your output concise.
                                                • Use simple vocabulary and straightforward syntax.}
==== USER PROMPT ====
Convert the following knowledge graph into a coherent English   ==== USER PROMPT ====
sentence or short paragraph.                    Convert the following knowledge graph into a coherent English sentence
Triples are given in the form ( subject | <P> predicate | <O>   or short paragraph.
object), separated by commas.                   Triples are given in the form ( subject | <P> predicate | <O> object),
                                                separated by commas.
Knowledge Graph:
( Arròs negre | <P> country | <O> Spain),    Knowledge Graph:
( Spain      | <P> ethnic Group | <O> Spaniards)   ( Arròs negre | <P> country | <O> Spain),
                                                ( Spain      | <P> ethnic Group | <O> Spaniards)
=== ASSISTANT (you) ===
<your generated text here>                       === ASSISTANT (you) ===
                                                <your generated text here>
```

Example Prompt

Figure 6: Zero-Shot Prompt Template for Knowledge Graph Verbalization Across Multiple LLMs

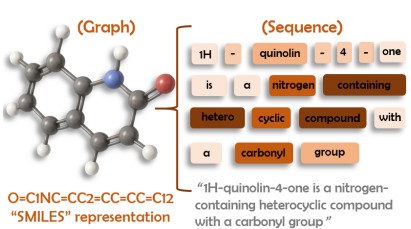 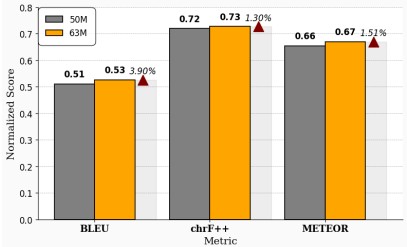

Figure 7: Comparison of (left) framing molecule captioning as a G2S task and (right) the performance of `DLM4G-1.o` and `DLM4G-2.o` models on the molecule captioning dataset.

### A.8 ZERO-SHOT PROMPTING

Zero-shot prompting (illustrated in Figure 6) exploits the rich, general-purpose knowledge encoded in pretrained large language models (LLMs) to tackle novel tasks without additional fine-tuning. By casting tasks as natural-language instructions or templated prompts, models such as GPT-3 Brown et al. (2020), DeepSeek Li et al. (2024b), LLaMa-3 Touvron et al. (2023), and Qwen2.5 Zeng et al. (2024) demonstrate strong out-of-the-box performance across diverse applications. Prior work has shown that LLMs internalize extensive linguistic, factual, and procedural knowledge during self-supervised training, yielding robust zero-shot capabilities in text classification Wang et al. (2022), machine translation Raffel et al. (2020), and code generation Chen et al. (2021a). A typical zero-shot prompt comprises three components:

1. A *system prompt* that assigns the model's role (e.g., "You are {MODEL}, a large language model. Convert RDF triples into fluent English.").

2. A *model-specific guidance* segment to steer style or brevity (e.g., "Keep your output concise.").

3. A *user prompt* presenting the task instance.

For example: *Convert the following knowledge graph into a single English sentence:*
$\langle S \rangle$ Arròs negre $\langle P \rangle$ country $\langle O \rangle$ Spain, $\langle S \rangle$ Spain $\langle P \rangle$ ethnic Group $\langle O \rangle$ Spaniards.

In this study, we evaluate four models—DeepSeek (7 B), GPT-o4-mini (8 B), LLaMa-3 (8 B), and Qwen2.5 (7 B)—to investigate how model scale, pretraining corpus, and architectural choices affect zero-shot generalization on knowledge-to-text tasks.

### A.9 MOLECULE CAPTIONING

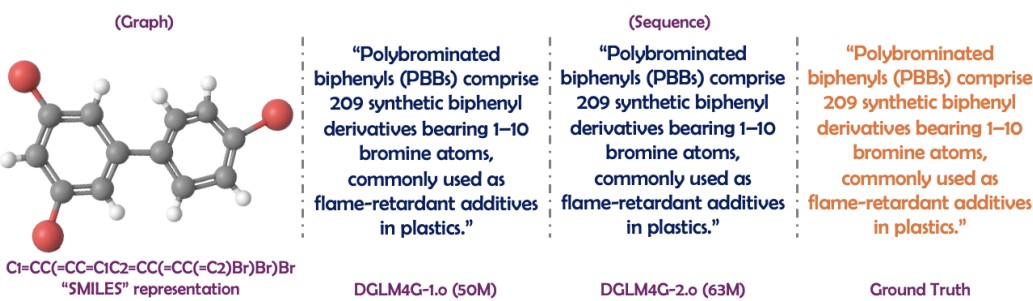

Figure 8: Qualitative Assessment of Molecule Captioning by `DLM4G` Given SMILES Representations

Figure 8 shows the captions produced by two variants of our model, `DLM4G-1.0` (50 M parameters) and `DLM4G-2.0` (63 M parameters), alongside the ground-truth description for a polybrominated biphenyl (PBB) molecule (SMILES shown beneath the 3D rendering). Both model outputs are nearly identical, correctly capturing: (1) The molecule class: "polybrominated biphenyls (PBBs) comprise 209 synthetic biphenyl derivatives", (2) The substitution range: "bearing 1–10 bromine atoms" and (3) The typical use case: "commonly used as flame-retardant additives in plastics."

Quantitatively, the two variants achieve very similar scores on all three evaluation metrics—BLEU, chrF++ and METEOR—reflecting their equivalently high factual fidelity and fluency. This example illustrates that even the smaller 50 M model matches the larger 63 M model in this task. Full dataset statistics and comprehensive metric results are provided in the main paper.

