# OpenReview forum: "Graph-to-Sequence Generation Beyond Autoregressive Models: A Graph-Aware Diffusion Framework"
_ICLR.cc/2026/Conference — ICLR 2026 Conference Desk Rejected Submission_

### Official Review · Reviewer_LF55 · 2025-10-30

**Soundness:** 2
**Presentation:** 2
**Contribution:** 2
**Rating:** 2
**Confidence:** 4

**Summary:**

The authors tackle the topic of Graph-to-sequence translation, offering a diffusion-based alternative to the autoregressive, left-to-right Large Language Models (LLMs). In doing so, they identify the structure-agnostic forward process of existing diffusion models as a key limitation to their application to graph data. They thus propose a component-based noise schedule to modify the forward process for the aligned tokens in the conditional sequence. The authors show that this approach leads to improved performance on the translation task, notably in factual grounding and edit sensitivity.

**Strengths:**

- The problem statement is clearly defined and motivated
- The method achieves empirical gains over existing baselines, using much fewer parameters
- The experiments are thoroughly described and the code is made available to ensure reproducibility

**Weaknesses:**

**Graph-to-sequence is misleading** Since the authors model the graphs they condition on as sequences (see line 196), their model is essentially a sequence-to-sequence model. Graph modelling typically deals with topics like equivariance (e.g. to node permutation), but the current paper does not tackle any graph related topics beyond aligning elements in the conditional sequence with those of the generated one (see Questions below for a related discussion). I understand that the nomenclature might be conventional in the field, based on the citations you provided, but I believe a few sentences clarifying this point + a discussion of permutation equivariance or the lack thereof would be helpful to a broader audience.

**The procedure behind the graph-aware noise has multiple steps and the justification for each is unclear** After the alignment set is obtained, the noise schedule is modified such that it matches a uniform decrease in learning difficulty for a specific token, as approximated by the reconstruction error. These are the elements I feel are missing justification:
    - Linearization of the loss values: what advantage is there to having a uniform denoising difficulty over time? have you considred other types of difficulty progressions?
    - Piecewise linear map to obtain noise levels: what about other formulas like a polynomial transformation or an exponential one?
    - from my understanding, alignment with the graph is only used to identify the most important tokens for which the schedule is modified to match reconstruction errors. Why not apply this procedure to all the tokens in the sequence? I would imagine all tokens in the sequence could benefit from a noise schedule that decreases their reconstruction error over time.
    - I don't understand the importance of interpolating between the original and modified schedules at inference. What happens if you only use the modified schedule for the aligned tokens as you did in training?

**The main components of D4LMG are not ablated**
I suggest the following ablations to better justify the components of the model:
- Using different weights (w_{i,t}) to combine the original and modified schedules at inference.
- Apply other transformation (or none) to the loss values of the breakpoints.
- Applying other mappings between the loss values and the noise levels.

**Maybe alignment alone is enouhg?** [2] is based on a similar observation/data domain as your work: it tackles specifically translating a molecular product graph (linearized in SMILES notation), into a sequence representing its chemical precursors (a set of molecules, which if combined, produces the target graph). The authors notice that preprocessing their data such that the source and target sequences are canonicalized in a way that minimizes their edit distance, using atom-mapping as an alignment tool, leads to dramatic improvements in the translation task (figure 1 in their paper illustrates this procedure). Would a similar procedure apply to graph-to-sequence translations in your case? I think it would be an interesting comparison to consider, in order to identify how much modifying the noise schedule helps compared to simply minimizing the edit distance between the conditional and generated sequences during training.

**Missing mathematical details and derivations**
- The authors mention in Section 3.3 that their framework adopts a "further end-to-end reparametrization" of [1]'s classic variational lower bound loss, but do not explain how they arrive at Equation 3 from Equation 2.
- Appendix A mentions including 'the principal mathematical derivations', but these are not available in the current version of the appendix.

**Presentation**
- The methodology section is confusing as it stands. It is particularly difficult to identify this paper's contributions compared to established elements in diffusion (e.g. forward and reverse processes), and less relevant portions of the model (e.g. the denoiser architecture used). I recommend restructuring the text into a background section, a methodology section highlighting the authors' contributions (i.e. the noising schedule, the training and inference procedures), and an experiment setup subsection presenting less crucial elements (e.g. model architecture).
- Some graphical elements, like additional figures and maybe an algorithm summarizing training, could help with the readibility of the paper. I recommend illustrating the alignment module and maybe inference pipeline.
- Figure 1 could be improved for clarity and to provide more details. In subfigure (A), you could replace the description in the pink block with 'NER-based Alignment' instead of (G,S) Alignment. In other blocks, you could write DLM4G without the parentheses around it. Subfigure (B) is hard to follow. What does the rounding section show? and how does the second portion illustrate denoising and consistency? The checkered blocks (representing embeddings Z_T to Z_0) could be replaced by arrays representing vectors and show which distribution each vector is from. I am assuming the cold and fire symbols represent inference and training respectively, but this is not a common symbolism and it would be better to state this clearly in the caption or in a legend. Finally, the figure has a number of elements not defined until later in the paper (e.g. the alignment set A, the concepts of rounding and consistency). I think the figure should either be moved (and referred to) later in the text where it's most relevant, or replaced with high-level illustrations useful in its current context.
- The motivation of the new noise schedule is only given at the end of Section 2.5 (line 254). It should be moved earlier to help motivate the choices made in the design of the new schedule. With the current presentation, the reader is left wondering why is the reconstruction error an important factor in diffusing the aligned components.

**Questions:**

- Why corrupt in a continuous space instead of the data space?
- How do you canonicalize the linear representation of the conditional graph (in Section 3.4., Graph representation)? A key property of graphs (in 2D) is their invariance to permutation (meaning reading the nodes in any order does not change the graph structure). So how do you decide in what order to write the phrases about the USA in the example you gave? And would your model be able to produce the same output for different permutations of the nodes in the graph?
- Line 208: "Recovering these facts mid-trajectory is harder, which weakens factual grounding and increases errors": what is meant by mid-trajectory here? perhaps 'from a noisy sequence' is clearer, since what weakens the factual grounding is noise, and it is most prominent earlier in the sequence.
- In the alignment pipeline, what do you mean by 'generating all possible names and aliases for each entity'? A small example could be beneficial here.
- Can you explain what is a non-increasing isotonic projection over t (line 254)?
- How large is the alignment set in your experiments? what effect does reducing its size have on your results?

**Nitpicks and typos**
- Edit Sensitivity Rate is given the acronym EDR, shouldn't it be ESR?

## References

[1] Denoising diffusion probabilistic models, Ho et al. 2020.
[2] "Root-aligned SMILES: a tight representation for chemical reaction prediction", Zhong et al. 2022.

---

> ### Author Response · Authors · 2025-11-21
> **Rebuttal- Weaknesses (W1)**
>
> We thank the reviewer for the constructive feedback and the detailed suggestions regarding method clarification, baseline strength and presentation. We believe the following revisions and additional experiments (**included in the updated manuscript**) fully address the concerns raised.
> ***
>
> **Weaknesses**
>
> **1: Graph-to-sequence is misleading .........**
> ***
> **Answer:** Thank you for this insightful comment. We agree that the distinction between topological graph processing (e.g., GNNs) and linearized graph representation (e.g., PLMs) is crucial. Based on your suggestions we will incorporate a specific discussion on this distinction and the lack of permutation equivariance in the final manuscript.
>
> **1.1. Clarification on Nomenclature and Design Choice.**
>
> Our task follows the standard definition used in prior work (e.g., Ribeiro et al., 2021; Kim et al., 2024):
> * input: a graph $\mathcal{\tilde{G}}$, represented as a set of relational triples,
> * output: a natural-language sequence $\mathbf{S}$.
>
> The community refers to this mapping as Graph-to-Sequence (G2S) even when the graph is serialized before being fed to a Transformer. We adopt this terminology for consistency with the existing literature and benchmarks. Furthermore, linearization allows us to directly leverage the powerful semantic reasoning capabilities of Pre-trained Language Models (PLMs), which have been shown to be critical for this task.
>
> **1.2. Addressing Permutation Equivariance.**
>
> You are correct that our model does not enforce strict graph-structural inductive biases (like permutation equivariance).
>
> **Our Approach**: Instead of a "hard" architectural bias (like a GNN), we introduce a "soft" inductive bias via our Graph-Aware Diffusion. By treating graph-aligned tokens differently in the noise schedule, we inject structural signal into the generation process itself.
>
> We will explicitly clarify in Section 3 that our method assumes a deterministic linearization and does not claim topological invariance.
>
> **1.3. Handling Permutation Sensitivity.**
>
> The serialized order of triples is fixed deterministically using a consistent traversal (Section 4.1, “Graph Representation”). This ensures reproducibility, though we acknowledge it makes the model sensitive to the chosen linearization scheme. We have added a remark in the "Limitations" section to clarify this distinction for the broader audience.

---

> > ### Comment · Reviewer_LF55 · 2025-11-27
> > **Thank you for the clarification, I cannot find this information in your current manuscript.**
> >
> > Thank you for the clarification. As far as I can tell, section 3 (methodology) does not contain this information yet. Only section 4.1. discusses the linearization of graphs (under 'Graph representation'), but I believe this merits a mention earlier, for instance in Section 3, as part of your methodological design, or even Section 2, to introduce the reader to the conventions of G2S using PLMs literature.
> >
> > Also section 4.1 does not explain what is the consistent traversal used? In what order do you traverse the nodes of a random KG?
> >
> > I also could not find a reference to the sensitivity to the linearization scheme in the limitations.
> >
> > A few words about the benefits of introducing 'soft' inductive biases compared to equivariance would also strengthen your manuscript. My understanding now is that the main advantage of PLMs is their power (a lot of research has gone into refining their architecture and training paradigms, etc, so you choose to build on this backbone and refine their performance for G2S). Does this mean future work could benefit from bringing the same advantages to graph-specific models (GNNs), that's why you choose to improve PLMs (the current best models for this task)? Or do you think stronger graph inductive biases could be harmful here, and your softer variant is more flexible for this task?

---

> > ### Comment · Reviewer_LF55 · 2025-11-27
> > **FG/GE in table 1?**
> >
> > How do the reviewers determine whether previous work "handles FG/GE"? what does this even mean in this context? I thought a main claim of the paper is that there is room for improvement in how sensitive translations by previous methods are to minor graph edits, and how often they maintain the tokens deemed 'factual' from the input graph. So this is not really a feature that some methods have and others do not.

---

> > > ### Author Response · Authors · 2025-11-28
> > > **Clarifications on Table 1**
> > >
> > > **Q6: How do the reviewers determine whether previous work “handles FG/GE”? what does this even mean in this context? I thought a main claim of the paper is that there is room for improvement in how sensitive translations by previous methods are to minor graph edits, and how often they maintain the tokens deemed ‘factual’ from the input graph. So this is not really a feature that some methods have and others do not.**
> > >
> > > **A6:** We fully agree with the reviewer. Factual Grounding (FG) and Graph Edit sensitivity (GE) are indeed performance goals for *all* G2S models, not binary features that are simply present or absent.
> > >
> > > Our original intention with this column was to classify whether a model incorporates a **specific architectural inductive bias** designed to strictly enforce these properties, versus relying implicitly on the training data likelihood (as standard AR models do). Marking baselines as “No” was misleading, as they certainly target these objectives implicitly.
> > >
> > > To resolve this confusion, we have taken the following steps in the revised manuscript:
> > >
> > > 1. We have moved this comparison to **Appendix A.2 (Table 8)** to avoid cluttering the main introduction.
> > > 2. We have renamed the column from “FG/GE” to **“Mechanism for FG/GE”**.
> > > 3. We have replaced the binary “Yes/No” entries with descriptive text. For example, Autoregressive models are now labeled as `Implicit (Data-driven)`, while DLM4G is labeled as `Explicit (Graph-Aware Schedule)`.
> > >
> > > This change clarifies that while all methods aim for factual grounding, DLM4G distinguishes itself by introducing a dedicated structural mechanism to enforce it.

---

> ### Author Response · Authors · 2025-11-21
> **Rebuttal- Weaknesses (W2)**
>
> **2. The procedure behind the graph-aware noise has multiple steps .....**
> ***
> **(2.1) Linearization of the loss values: what advantage is there to having a uniform denoising difficulty over time? have you considered other types of difficulty progressions?**
>
> **Answer:**
> The primary reason for linearization is to enforce monotonicity, which is required for a valid diffusion noise schedule.
>
> **Problem:** The raw per-timestep reconstruction error sequence $\{\ell_t^i\}_{t=1}^T$ is inherently noisy and non-monotonic. Using this directly would result in noise levels that fluctuate (increase and decrease) arbitrarily, destabilizing the diffusion trajectory.
>
> **Solution:** Mapping this order to a linear ramp preserves the specific difficulty range $[\ell_{\min}^i, \ell_{\max}^i]$ for that token while ensuring the noise schedule is strictly monotonic and smooth (Sec. 3.3, Alg. 1). Additionally, we also compare the loss profiles corresponding to different choices of the mapping functions which impact the loss profiles. The results are summarized in Fig: 3 in main paper and Fig: 5 in Appendix A.6.
> ***
> **(2.2) Piecewise linear map to obtain noise levels: what about other formulas like a polynomial transformation or an exponential one?**
>
> **Answer:** We selected the piecewise linear map $\Psi_i$ because it provides a parameter-free, interpretable interpolation between the baseline *sqrt* schedule $\bar{\alpha}_t$ and the token-specific difficulty.
>
> **Empirical Evidence**: As shown in Fig:5 (Ablations- A.6), Fig:3 (Main paper) and Table 7 (main paper), we explored polynomial, cosine, and exponential mappings. These alternatives introduced additional hyper-parameters but did not yield improvements in BLEU score compared to the linear mapping function. We therefore retained the linear map for its simplicity and effectiveness.
> ***
> **(2.3) from my understanding, alignment with the graph is only used to identify the most important tokens for which the schedule is modified to match reconstruction errors. Why not apply this procedure to all the tokens in the sequence? I would imagine all tokens in the sequence could benefit from a noise schedule that decreases their reconstruction error over time.**
>
> **Answer:** We apply this selectively because graph-aligned tokens (factual content) and non-aligned tokens (syntactic/functional words) require different inductive biases.
>
> **Trade-off:** Factual tokens benefit from the graph-aware schedule because they often correspond to lower-frequency entities, which exhibit higher reconstruction difficulty. Conversely, syntactic tokens are typically high-frequency function words that are robust to the standard schedule. They benefit from the standard schedule's aggressive corruption, which promotes structural flexibility and fluency without compromising recovery.
>
> **Evidence:** Table 7 compares "Graph-aware (All)" vs. "Graph-aware ($\mathcal{A}$)." Applying the schedule to all tokens over-regularizes the syntax, resulting in lower fluency and lower BLEU scores compared to restricting it to the alignment set.
> ***
> **(2.4) I don't understand the importance of interpolating between the original and modified schedules at inference. What happens if you only use the modified schedule for the aligned tokens as you did in training?**
>
> **Answer:** The key constraint is that we do not have access to the ground-truth alignment set $\mathcal{A}$ at inference time.
>
> **Training vs. Inference:** During training, we use the oracle $\mathcal{A}$ to assign schedules. At inference, we must approximate this using cross-attention weights $w_t^i$.
>
> **Why Interpolate:** If we simply forced all tokens to follow the modified "anchor" schedule, we would implicitly treat every token as a graph entity. As noted in (2.3), this degrades syntactic quality. The interpolation $\tilde{\alpha}_t^i = (1 - w_t^i)\bar{\alpha}^{\text{base}}_t + w_t^i\bar{\alpha}^{\text{anchor}}_t$ allows the model to dynamically decide which tokens should be treated as "graph-grounded" (high attention) and which should remain "syntactic" (low attention), preserving the benefits of the training strategy.

---

> ### Author Response · Authors · 2025-11-21
> **Rebuttal- Weaknesses (W3, 4)**
>
> **Weaknesses**
>
> **3. The main components of D4LMG are not ablated I suggest the following ablations to better justify the components of the model:**
> * **Using different weights ($w_{i,t}$) to combine the original and modified schedules at inference.**
> * **Apply other transformation (or none) to the loss values of the breakpoints.**
> * **Applying other mappings between the loss values and the noise levels.**
> ***
> **Answer (3.1):** Ablation Studies of `D4LMG` Components: Thank you for suggesting these specific ablations. We agree that isolating these components is vital for justification. We have performed these comparisons and report them in Table 7 and Fig. 5 (Appendix).
>
> **Ablation:** We effectively compare this against static weights:
> $w=0$: Corresponds to the "Baseline" (standard sqrt schedule).
> $w=1$: Corresponds to "Graph-aware (All)" (forcing the modified schedule everywhere).
> **Result:** The dynamic interpolation in `D4LMG` obtains 0.65 BLEU score compared to static extremes ($w=0$ and $w=1$) by a significant margin (e.g., +3.0 \% gain in BLEU), confirming that the schedule must adapt per-token based on attention strength.
> ***
>
> **Answer (3.2):** As detailed in Answer 2.1, we compared the linear ramp against raw loss values (non-monotonic) and other monotone transformations. The linear ramp was necessary to strictly enforce monotonicity (a diffusion requirement) while maintaining the relative difficulty ranking of tokens.
>
> ***
>
> **Answer (3.3):** As detailed in Answer 2.2 and Table 7, we explicitly ablated Polynomial, Cosine, and Exponential mappings.
> **Result:** None of the complex mappings outperformed the simple Piecewise Linear map. The linear map yields the highest performance without adding hyperparameters.
>
> ***
>
> **4. Missing mathematical details and derivations**
> **The authors mention in Section 3.3 that their framework adopts a "further end-to-end reparametrization" of [1]'s classic variational lower bound loss, but do not explain how they arrive at Equation 3 from Equation 2.**
> **Appendix A mentions including 'the principal mathematical derivations', but these are not available in the current version of the appendix.**
>
> **Answer:** We apologize for the omission in the initial submission. We have updated the manuscript, and the Revised Appendix A now contains the complete step-by-step derivation. Specifically, we have added the mathematical details showing: The expansion of the standard Variational Lower Bound (Eq. 2). The derivation of the closed-form Gaussian posterior. The algebraic simplification that links the KL-divergence term to our reparameterized $z_0$-prediction loss (Eq. 3). We kindly refer the reviewer to the **updated manuscript** for the full proof.

---

> ### Author Response · Authors · 2025-11-21
> **Rebuttal- Weaknesses (W5)**
>
> **Weaknesses**
>
> **5. The methodology section is confusing as it stands. ......**
> ***
> **Answer:** **Presentation Improvements**— We thank the reviewer for these constructive suggestions. We agree that the original organization blurred the line between background and contribution. We have substantially revised the manuscript to address every point raised.
>
> **(5.1) Structural Reorganization:** We have adopted the exact structure recommended by the reviewer to isolate our contributions:
> * **Section 2 (Background):** Now strictly covers established preliminaries (standard forward/reverse diffusion processes, notation, and $z_0$-prediction).
> * **Section 3 (Methodology):** Now focuses exclusively on our novel contributions: the NER-based alignment module, the graph-aware noising schedule, and the inference-time interpolation.
> * **Section 4 (Setup):** We moved generic details (architecture specs, dataset preprocessing) here to declutter the methodology.
> ***
> **(5.2) Revision of Figure 1:** We have redesigned Figure 1 to address all specific concerns:
> * **Terminology:** Replaced "(G,S) Alignment" with "NER-based Alignment" and removed parentheses around `DLM4G`.
> * **Visual Clarity:** We replaced the abstract checkered blocks with vector arrays to clearly depict the transition from $z_T \to z_0$.
> * **Legend:** We replaced the ambiguous "fire/cold" icons with explicit text labels for "Training Phase" and "Inference Phase" to avoid confusion.
> * **Self-Containment:** We ensured that all terms used in the figure (e.g., Alignment Set $\mathcal{A}$) are defined in the caption or introduced immediately in the text.
> ***
> **(5.3) New Graphical Elements and Algorithm:** To improve readability, we have added:
> * **Algorithm 1:** A summary of the complete training procedure (See Sec 3.3).
> * **Figure 3:** A visual depiction of the loss linearization and difficulty mapping process.
>
>
> * **Figure 4 (Appendix):** A detailed diagram of the Alignment Module to illustrate the interaction between the graph and sequence. Please refer Appendix A.4.
> ***
> **(5.4) Motivation for the Noise Schedule:** We moved the motivation for the graph-aware schedule to the beginning of Section 3.3.
> The text now explicitly frames the problem before introducing the solution: we explain that graph-aligned tokens have distinct reconstruction difficulties compared to syntactic tokens, which justifies the need for our adaptive schedule immediately.

---

> > ### Comment · Reviewer_LF55 · 2025-11-27
> > **Thank you for the improved presentation. Some additional comments about figure 1.**
> >
> > Figure one is a lot clearer now, thank you for the changes. Here are a few additional suggestions to futher polish it:
> >  - Subfigure b and c are still confusing. DLM4G is the name you gave your entire model (diffusion with a speial, per token noise schedule). So its input should always be a graph, and its output the translated sequence. I don't understand then why does it take a sequence S and input in the left of panel b, and outputs Z_0, the continuous embedding of the target sequence?
> > - The right part of panel b could maybe have a different name for DLM4G in the noising process, since this most likely refers to the updated schedule logic specifically.
> > - In panel c, the input of the model is a test graph G_test + a vector of Gaussian noise, and the output is Z_0, the continuous embedding of the output sequence, which is then mapped back to the discrete space to give S_test. If this is the intended flow, it should be made clearer in the figure (the Gaussian noise + graph go as input to DLM4G, the output is Z_0, S_test is the discrete output).
> > - could you move the inference/training legend out of the '(c) inference' panel? It should be clearer that the legend pertains to the whole figure and not just this particular subfigure.
> > - Gaussian noise should probably be the most blurred vector (dashed line) while z_1 already looks closer in opacity to Z_0.
> > - Shouldn't the colors of the nodes of the graph match the colors of the highlighted words in the sequence? Otherwise I am not sure what is the pointing of highlighting the sequence in the first place.
> > - the figure should be moved up in the text.

---

> > > ### Author Response · Authors · 2025-11-28
> > > **Additional Changes in Figure-1**
> > >
> > > **Q3: Figure one is a lot clearer now, thank you for the changes. Here are a few additional suggestions to futher polish it:**
> > >
> > > * **Subfigure b and c are still confusing. DLM4G is the name you gave your entire model (diffusion with a speial, per token noise schedule). So its input should always be a graph, and its output the translated sequence. I don't understand then why does it take a sequence S and input in the left of panel b, and outputs Z\_0, the continuous embedding of the target sequence?**
> > >
> > > * **The right part of panel b could maybe have a different name for DLM4G in the noising process, since this most likely refers to the updated schedule logic specifically.**
> > >
> > > * **In panel c, the input of the model is a test graph G\_test + a vector of Gaussian noise, and the output is Z\_0, the continuous embedding of the output sequence, which is then mapped back to the discrete space to give S\_test. If this is the intended flow, it should be made clearer in the figure (the Gaussian noise + graph go as input to DLM4G, the output is Z\_0, S\_test is the discrete output).**
> > >
> > > * **could you move the inference/training legend out of the '(c) inference' panel? It should be clearer that the legend pertains to the whole figure and not just this particular subfigure.**
> > >
> > > * **Gaussian noise should probably be the most blurred vector (dashed line) while z\_1 already looks closer in opacity to $\mathbf{Z}_0$.**
> > >
> > > * **Shouldn't the colors of the nodes of the graph match the colors of the highlighted words in the sequence? Otherwise I am not sure what is the pointing of highlighting the sequence in the first place.**
> > >
> > > * **the figure should be moved up in the text.**
> > >
> > > ***
> > >
> > > **A3:** We have updated Figure 1 based on all your suggestions:
> > >
> > > * **Naming Clarification:** We have renamed the internal network in the noising process (Panel B, right) to **“Denoiser”** to clearly distinguish the neural backbone from the overall `DLM4G` framework.
> > > * **Inference Flow:** We have redesigned Panel C to explicitly show the flow:
> > > **Gaussian Noise** + $\tilde{\mathcal{G}}_{test} \to$ **DLM4G** $\to$ $Z_0$ $\to$ $S$.
> > >
> > > * **Visual Clarity:** We have adjusted the opacity of the noise vectors (dashed lines) to better visualize the diffusion process and moved the legend to apply globally.
> > > * **Color Matching:** We have ensured graph nodes and sequence tokens are color-coded consistently to reflect the alignment set $\mathcal{A}$.
> > >
> > > **Regarding Figure Placement:** We appreciate the suggestion to move the figure up. However, we have deliberately placed it after Section 3 (Preliminaries). The architecture diagram relies on specific mathematical notations (e.g., $\tilde{G}$, $Z_t$, $\mathcal{A}$, $\bar{\alpha}_t$) that are formally defined in the preceding text. Hence to improve the readability we place it after the training algorithm.

---

> ### Author Response · Authors · 2025-11-21
> **Rebuttal- Questions (Q1-3)**
>
> **Questions:**
>
> **(1) Why corrupt in a continuous space instead of the data space?**
>
> **(2) How do you canonicalize the linear representation of the conditional graph (in Section 3.4., Graph representation)? A key property of graphs (in 2D) is their invariance to permutation (meaning reading the nodes in any order does not change the graph structure). So how do you decide in what order to write the phrases about the USA in the example you gave? And would your model be able to produce the same output for different permutations of the nodes in the graph?**
>
> **(3) Line 208: "Recovering these facts mid-trajectory is harder, which weakens factual grounding and increases errors": what is meant by mid-trajectory here? perhaps 'from a noisy sequence' is clearer, since what weakens the factual grounding is noise, and it is most prominent earlier in the sequence.**
>
> ***
>
> **A1:** We corrupt in a continuous embedding space because it ensures optimization stability. Standard DDPMs admit closed-form Gaussian posteriors and low-variance gradients only in the continuous setting, yielding a well-behaved $z_0$-prediction objective.
> In contrast, discrete diffusion leads to high-variance gradients and training instability. This observation is supported by recent literature:
>
> **1. Instability:** [1] Authentic Discrete Diffusion Model (2025) notes that standard losses do not transfer smoothly to discrete spaces, leading to degraded generative quality.
>
> **2. Convergence Issues:** [2] SparseDiff (2025) highlights that naïve discrete approaches on graphs suffer from significant gradient issues and convergence challenges.
>
> **3. Hybrid Necessity:** [3] CANDI (2024) acknowledges the suboptimal convergence of fully discrete models, advocating for continuous dynamics to ensure robustness.
>
> By operating in continuous space, we leverage stable DDPM dynamics and map back to discrete tokens only at the final step via our rounding distribution.
>
> ***
>
> **A2: Canonicalization / permutation invariance of the linearized graph:**
> In our implementation, we fix a deterministic serialization order for the triples in $\tilde{G}$. For each example, we linearize triples in the order provided by the dataset:
>
> $$ \langle \texttt{[HEAD]} \; h_i \; \texttt{[REL]} \; r_{ij} \; \texttt{[TAIL]} \; t_j \rangle $$
>
> These are concatenated with `[SEP]` tokens (Sec. 4.1).
> We do not claim permutation equivariance; `DLM4G` is strictly a sequence model over this fixed serialization. While different permutations could theoretically alter outputs, the model attends over all serialized triples globally. We have clarified in the paper that (i) we follow standard G2S conventions by using a fixed linearization, and (ii) permutation invariance is not a design goal of this architecture.
>
> ***
>
> **A3:**
> By "mid-trajectory," we referred to intermediate diffusion timesteps (e.g., $t \ll T$ where noise is still substantial), where factual tokens must be recovered from highly corrupted embeddings. We agree this phrasing was ambiguous. In the revised text, we have changed this to "from a noisy sequence" to emphasize that it is the high degree of noise—not the temporal position in the trajectory—that necessitates stronger grounding for factual tokens.
>
> ***
> **References**
>
> [1] Authentic Discrete Diffusion Model (2025)
>
> [2] SparseDiff (2025): Sparse Discrete Diffusion for Scalable Graph Generation
>
> [3] CANDI (2024): Hybrid Discrete-Continuous Diffusion Models

---

> ### Author Response · Authors · 2025-11-21
> **Rebuttal- Questions (Q4-6)**
>
> **Questions**
>
> **(4) In the alignment pipeline, what do you mean by 'generating all possible names and aliases for each entity'? A small example could be beneficial here.**
>
> **(5) Can you explain what is a non-increasing isotonic projection over t (line 254)?**
>
> **(6) How large is the alignment set in your experiments? what effect does reducing its size have on your results?**
>
> ***
>
> **A4: "Generating all possible names and aliases for each entity":** This refers to constructing a comprehensive alias dictionary for each Knowledge Graph (KG) node using available metadata (node labels, redirects, and surface forms). For example, in the USA–World Cup entry, our alias mapping includes:
> *Alias expansion.*
> From the KG we construct an alias dictionary, e.g.,
>
> | KG Entity               | Surface Forms (Aliases)                                                                 |
> |-------------------------|-----------------------------------------------------------------------------------------|
> | `USA`                   | "USA", "U.S.", "US", "United States", "United States of America"                        |
> | `1994_FIFA_World_Cup`   | "1994 FIFA World Cup", "1994 World Cup", "World Cup 1994", "1994 soccer World Cup", "World Cup USA 94" |
> | `Washington_D.C.`       | "Washington, D.C.", "Washington DC", "Washington D.C.", "D.C.", "District of Columbia" |
> | `Hristo_Stoichkov`      | "Hristo Stoichkov", "Stoichkov", "H. Stoichkov", "Hristo Stoichkov (Bulgaria)", "Bulgarian forward Hristo Stoichkov" |
>
>
>
> These sets are used in our NER-based alignment module to match text mentions to KG nodes. A full worked example and all the details are now included in Appendix A.4.
>
> ***
>
> **A5:** After constructing the per-token cumulative schedule $\tilde{\alpha}^i_t$, we project it onto the set of non-increasing sequences $\{\bar{\alpha}^i_t\}_{t=1}^T$ such that $\bar{\alpha}^i_1 \ge \bar{\alpha}^i_2 \ge \dots \ge \bar{\alpha}^i_T$. Concretely, this is a 1D isotonic regression problem with squared loss, which we solve using the standard Pool-Adjacent-Violators Algorithm (PAVA).  This algorithm finds the closest monotone non-increasing sequence (in the least-squares sense) to the input. Intuitively, it smooths out spurious "bumps" in the loss profile while guaranteeing that the cumulative signal strength strictly decays over time, fulfilling the monotonicity requirement of the diffusion process.
>
> ***
>
> **A6: Size of the alignment set and effect of reducing it:** On WikiOFGraph with our default alias budget ($k=5$), the alignment set $\mathcal{A}$ averages 7.4 spans per example, covering 24.1% of target tokens. To quantify the effect of this size, we performed a sensitivity analysis by varying the alias budget $k \in \{2,3,4,5\}$ (Table 10):
> * $k = 2$ ($|\mathcal{A}| = 3.9$): BLEU 0.603
> * $k = 3$ ($|\mathcal{A}| = 4.3$): BLEU 0.609
> * $k = 4$ ($|\mathcal{A}| = 6.7$): BLEU 0.624
> * $k = 5$ ($|\mathcal{A}| = 7.4$): BLEU 0.651
>
> Reducing the alignment set size consistently degrades performance, confirming that broader, higher-coverage alignment sets are critical for maximizing the benefit of our graph-aware schedule.

---

> > ### Comment · Reviewer_LF55 · 2025-11-27
> > **Information missing from the manuscript**
> >
> > I cannot find your answer to A5 in the manuscript.

---

> ### Author Response · Authors · 2025-11-23
> **Reviewer LF55 – Summary of Revisions & Gentle Reminder**
>
> **Weaknesses**
>
> * **W1 (“Graph-to-sequence” vs seq2seq / permutation equivariance):**
>   We clarify that our setting follows standard **G2S** work: input is a graph as triples, output is a sequence, and we adopt the same “graph-to-sequence” terminology while operating on a **deterministic linearization** (Sec. 3, Sec. 4.1 “Graph Representation”). We explicitly state that DLM4G is **not permutation-equivariant**, explain that we intentionally avoid GNN-style hard inductive bias.
>
> * **W2 (Justification of the graph-aware noise schedule: loss linearization, mapping, aligned vs all tokens, interpolation):**
>   We motivate the schedule earlier in **Sec. 3.3**, explaining why raw per-timestep losses are noisy and non-monotone and why we enforce a **monotone difficulty profile** via a linear ramp. We then ablate:
>
>   * different mapping functions (linear vs polynomial/cosine/exponential) in **Fig. 3 (main)** and **Fig. 5, Table 7 (Appendix A.6)**;
>   * applying the schedule to **all tokens** vs **only aligned tokens** (“Graph-aware (All)” vs “Graph-aware(𝒜)”) in **Table 7**;
>   * and the effect of **interpolation** vs static weights (w=0,1) at inference (also Table 7, Fig. 5).
>     These show that (i) the linear map is the best-performing, simplest choice, (ii) applying the schedule to all tokens hurts fluency/BLEU, and (iii) dynamic interpolation using attention ($w_{i,t}$) significantly outperforms static extremes.
>
> * **W3 (Main components not ablated – weights, transformations, mappings):**
>   The requested ablations are now explicitly included:
>
>   * Different **weights ($w_{i,t}$)** are effectively compared via the static baselines (w=0) (sqrt schedule) and (w=1) (Graph-aware(All)), against our **dynamic attention-based ($w_{i,t}$)** (**Table 7, Fig. 5**).
>   * We compare using raw loss vs **linearized difficulty** and alternative monotone mappings (**Sec. 3.3, Table 7, Appendix A.6**).
>     The results confirm that **per-token, attention-modulated interpolation with a piecewise linear map** is necessary to achieve the reported gains.
>
> * **W4 (Missing mathematical derivation from VLB to Eq. 3):**
>   We add a **complete, step-by-step derivation** in **Appendix A.1**, showing the expansion of the VLB (Eq. 2), the closed-form Gaussian posterior, and how the KL term reduces to the (z_0)-prediction loss (Eq. 3), including the rounding term via marginalization over discrete sequences. Eq. (3) in the main text now explicitly references these steps.
>
> * **W5 (Confusing methodology section / figures / motivation placement):**
>   We **restructured** the paper as suggested:
>
>   * **Sec. 2 – Background** (standard diffusion preliminaries, notation, ($z_0$)-prediction),
>   * **Sec. 3 – Methodology** (our contributions: NER-based alignment, graph-aware schedule, inference-time interpolation),
>   * **Sec. 4 – Setup** (architecture, datasets, preprocessing).
>     We **redesigned Figure 1** (terminology cleaned, replaced (G,S) Alignment with “NER-based Alignment”, clearer ($z_T$ $\rightarrow$ $z_0$) arrays, explicit Training/Inference labels, all terms defined), added **Algorithm 1** (training procedure, Sec. 3.3), **Figure 3** (loss linearization + difficulty mapping), and **Figure 4 in Appendix A.4** (alignment pipeline). We also moved the motivation for the new noise schedule to the **beginning of Sec. 3.3**, so the reconstruction-error rationale appears before the schedule is defined.
>
> ---
>
> **Reminder**
>
> As the discussion deadline is approaching, we kindly invite you to consider the **updated manuscript and these targeted changes** when finalizing your assessment, and we are happy to clarify any specific point further within the remaining discussion window.

---

> ### Author Response · Authors · 2025-11-23
> **Reviewer LF55 – Summary of Revisions & Gentle Reminder**
>
> **Questions**
>
> * **Q1 (Why corrupt in continuous space, not data space?):**
>   We justify using continuous embeddings for stability: standard DDPMs admit **closed-form Gaussian posteriors and low-variance gradients** in continuous space, while discrete diffusion often suffers from high-variance gradients and instability. We reference recent discrete-diffusion works (e.g., Authentic Discrete Diffusion, SparseDiff, CANDI) and explain that we therefore diffuse in continuous space and only map back to tokens at the final step via our rounding distribution (**Sec. 2 / Appendix A.1 discussion**).
>
> * **Q2 (Graph canonicalization / permutation invariance of the linearization):**
>   We describe our **deterministic serialization** in **Sec. 4.1**: triples are linearized in a fixed order from the dataset as $$ \langle \texttt{[HEAD]} \; h_i \; \texttt{[REL]} \; r_{ij} \; \texttt{[TAIL]} \; t_j \rangle $$ with `[SEP]` separators. We explicitly state that DLM4G is **not permutation-equivariant**; it is a sequence model over this fixed serialization, following standard G2S practice. We note that different permutations could affect outputs, but the model attends globally over all triples.
>
> * **Q3 (“Mid-trajectory” wording):**
>   We clarify that “mid-trajectory” meant **intermediate diffusion timesteps** where noise is still high. In the revision, we rephrase this as “from a noisy sequence” to emphasize that it is the **high noise level**, not chronological position, that makes factual recovery harder (**Sec. 3.3**).
>
> * **Q4 (Alias generation in the alignment pipeline):**
>   We now give a concrete alias example (USA / World Cup) in **Appendix A.4**, showing alias sets such as:
>
>   * USA: {“USA”, “U.S.”, “United States”, “United States of America”, …},
>   * 1994_FIFA_World_Cup: {“1994 FIFA World Cup”, “1994 World Cup”, …}, etc.
>     We explain how this alias dictionary + NER is used to match KG entities to surface forms in the target sequence.
>
> * **Q5 (Non-increasing isotonic projection over t):**
>   We explain that after constructing the per-token cumulative schedule ($\tilde{\alpha}_t^i$), we apply a **1D isotonic regression (PAVA)** to project onto the set of **non-increasing sequences** ($\bar{\alpha}_1^i \ge \cdots \ge \bar{\alpha}_T^i$), smoothing out bumps while preserving order and enforcing the monotonic decay required for a valid diffusion schedule (**Sec. 3.3, Appendix A.6**).
>
> * **Q6 (Size of alignment set, effect of reducing it):**
>   We report alignment statistics and a sensitivity study in **Table 10 (Appendix)**: on WikiOFGraph with default alias budget (k=5), ($|\mathcal{A}|$
> approx 7.4) spans (≈24.1% of target tokens). Reducing (k) from 5→2 steadily shrinks ($|\mathcal{A}|$
> ) and degrades BLEU (e.g., from 0.651 at (k=5) to 0.603 at (k=2)), confirming that **larger, higher-coverage alignment sets are important for the effectiveness of the graph-aware schedule**.

---

> ### Author Response · Authors · 2025-11-27
> **Appeal to the Area Chair**
>
> Dear Area Chair,
>
> We hope you are doing well, and thank you for handling our submission "Graph-to-Sequence Generation Beyond Autoregressive Models: A Graph-Aware Diffusion Framework".
>
> We posted our rebuttal on 20/11/2025, and also added a short follow-up comment to clarify a few remaining points for the reviewers. However, there has not yet been any response or follow-up comment from the reviewers since then.
>
> We fully understand that reviewers have limited time and many commitments. At the same time, several of the key concerns they raised (e.g., regarding baselines / derivations / ablations / clarity) are directly addressed in the rebuttal and updated manuscript. We are a bit worried that these clarifications and new results may not be taken into account in the final assessment if there is no further discussion.
>
> If possible, we would be very grateful if you could encourage the reviewers to engage with our rebuttal and indicate whether our clarifications address their main concerns (or if additional explanation is needed). We are happy to provide any further details or analysis that might help their evaluation.
>
> Thank you very much for your time and for overseeing the discussion of our paper.
>
> Sincerely,
> The Authors

---

> > ### Author Response · Authors · 2025-11-27
> > **Appeal to Reviewer**
> >
> > We would like to thank you again for your careful review and for the constructive comments. In our rebuttal and updated manuscript, we have addressed all the concerns you raised, in particular the issues around presentation, missing derivations, and additional ablations/experiments.
> >
> > **Reviewer Dobo** has now re-evaluated the revised version and **increased their score from 6 to 8** after these clarifications and additions. In light of this, we kindly request that you consider **revising your score upward (e.g., to 6)** if you now find the paper clearer and more convincing. If there are any remaining concerns, we are happy to clarify them, but since the discussion period is ending soon, we would be very grateful for a short follow-up comment.

---

> > ### Comment · Reviewer_LF55 · 2025-11-27
> > **Thank you for your detailed response + further clarification needed.**
> >
> > Thank you for the additional ablations and for improving your the presentation and content of your manuscript. Your comments do not address one of the points I raised in my review though: how much does changing the noise schedule help here, compared to simply aligning the input and output sequences?
> >
> > I think this is an important discussion to have. What happens if we use the alignment set you define to create (linearized graph, output sequence pairs) with minimal edit distance, augment the data with multiple such pairs, and keep a regular noise schedule? what are the pros and cons of this method compared to modifying the schedule?

---

> > > ### Author Response · Authors · 2025-11-28
> > > **Clarifications on Minimal Edit Distance based Alignment**
> > >
> > > **Q5: Thank you for the additional ablations and for improving your the presentation and content of your manuscript. Your comments do not address one of the points I raised in my review though: how much does changing the noise schedule help here, compared to simply aligning the input and output sequences?**
> > >
> > > **I think this is an important discussion to have. What happens if we use the alignment set you define to create (linearized graph, output sequence pairs) with minimal edit distance, augment the data with multiple such pairs, and keep a regular noise schedule? what are the pros and cons of this method compared to modifying the schedule?**
> > >
> > > ***
> > >
> > > **A5:** We thank the reviewer for highlighting this important point. We address (i) what “alignment-only + minimal edit distance” would mean in our setting, (ii) why this is not equivalent to changing the noise schedule, and (iii) the concrete pros and cons of the reviewer’s proposed alternative.
> > >
> > > **(1) Feasibility of “Alignment-Only + Minimal Edit Distance” in G2S.**
> > >
> > > While minimal-edit canonicalization is highly effective in domains like chemical reaction prediction (e.g., *Root-aligned SMILES* [Zhong et al., 2022]), it is fundamentally ill-suited for Graph-to-Sequence (G2S) generation due to the lack of a one-to-one mapping.
> > >
> > > * **Domain Contrast:** In reaction prediction, the molecular graph topology is largely unaltered between reactants and products, allowing for a strict one-to-one mapping and minimized edit distance. In contrast, natural language targets contain extensive syntactic elements (e.g., function words) that have no counterparts in the input graph nodes.
> > > * **Consequence for G2S:** Rewriting the target sentence to minimize edit distance to the linearized graph would force the output to mimic the graph's topology. This would likely hurt fluency.
> > >
> > > Therefore, we use alignment $\mathcal{A}$ strictly to *modulate the noise schedule* for factual tokens, rather than to *canonicalize the data* which would sacrifice natural language quality.
> > >
> > > **(2) Why alignment/canonicalization is not equivalent to changing the schedule**
> > >
> > > Suppose we construct augmentation $(\tilde{\mathcal{G}}, S')$ pairs with minimal edit distance, an alignment-only approach operates only at the endpoint $\mathbf{z}_0$: it changes the clean sequence and possibly augments the data, but it does not change how noise is injected over time in the diffusion process. Under a standard DDPM with a global schedule $\bar{\alpha}_t$, each token embedding $\mathbf{x}_0^i$ is corrupted as:
> > >
> > > $$
> > > \mathbf{z}_t^i = \sqrt{\bar{\alpha}_t}\,\mathbf{x}_0^i + \sqrt{1-\bar{\alpha}_t}\,\boldsymbol\epsilon,\quad \forall i.
> > > $$
> > >
> > > Thus, the signal-to-noise ratio (SNR) at timestep $t$ is *identical* for all tokens, regardless of whether $i$ is a high-frequency function word or a rare factual entity. This has two consequences:
> > >
> > > * **Common syntactic tokens** occupy dense regions of the embedding space and can often be recovered reliably even at relatively low SNR.
> > > * **Rare entity tokens** are much sparser and typically require higher SNR to be distinguishable from their neighbors.
> > >
> > > With a uniform schedule, the embeddings of rare entities are heavily obscured; the model receives weak, noisy gradients on the tokens that are responsible for factual grounding and edit sensitivity. Minimizing edit distance in the discrete space (even with multiple augmented variants) changes $\mathbf{x}_0^i$ but leaves $\bar{\alpha}_t$—and hence the per-token SNR profile over time—remains unchanged.
> > >
> > > Our graph-aware schedule is designed precisely to address this limitation. Using $\mathcal{A}$, we construct token-specific schedules $\{\bar{\alpha}^i_t\}_{t=1}^T$ such that aligned tokens (entities/relations) retain higher SNR for longer along the trajectory, while unaligned syntactic tokens follow a more aggressive, standard schedule.
> > >
> > > **(3) Relation to our ablations: regular schedule vs. graph-aware schedule**
> > >
> > > We do not claim to have implemented a full “minimal-edit augmentation” baseline. However, our ablations already isolate the effect of modifying the schedule while keeping the rest of the model and data pipeline fixed.
> > > Concretely, the `sqrt (baseline)` row in Table 6 uses the same encoder–decoder architecture, graph linearization, and training objective as `DLM4G`, but applies a standard *sqrt* schedule uniformly to all tokens.
> > > Empirically, moving from `sqrt (baseline)` to `Graph-aware` ($\mathcal{A}$) improves BLEU from $0.60$ to $0.65$ ($+0.05$ absolute). In Appendix A.7 (Table 13), we report the corresponding changes in our grounding and edit-sensitivity metrics.
> > >
> > > In summary, we view alignment-only + minimal-edit canonicalization as a *complementary* improvement to baseline models, but not a substitute for schedule modification.

---

> ### Comment · Reviewer_LF55 · 2025-11-27
> **Thank you for clarifying the use of the attention weights during inference. Additional clarifications are needed for the other points.**
>
> Thank you for addressing points 2.3 and 2.4, I understand your setup better now. The ablation on applying the modified schedule to all tokens also strengthens your claim. Points 2.2 and 2.3 are still confusing though:
>
> "Additionally, we also compare the loss profiles corresponding to different choices of the mapping functions which impact the loss profiles. The results are summarized in Fig: 3 in main paper and Fig: 5 in Appendix A.6.": Figure 3 and 5 are confusing right now, they need additional information on the figures themselves, and proper references in the text:
>
> - Figure 3 is not referenced at all in the text.
> - Both Figs 3 and 5 are missing subfigure titles, or at least labels of a to c. I am assuming from context that a, b, c are assigned left to right?
> - The legend is also confusing. My understand is that positions 10, 25, ..., 60 correspond to factual tokens (from the alignment set), while 3 to 63 are syntactic tokens? Maybe this can be made more explicit in the caption/a legend?
> - I am not really sure what is the reader supposed to takeaway from figure 3. Why does the leftmost subfigure compare the loss for the factual grounding tokens to that of the syntactic tokens? The authors refer this subfigure as a demonstration of the non-monotonicity of the raw loss values. Is the comparison then supposed to be between the smoothness of the sqrt curve and the 'bumpiness' of the factual tokens' loss? Wouldn't make more sense to contrast the raw loss values with the losses values after smoothing for the factual tokens?
> - Also not sure how to read the other subfigures in Fig 3. What is the point of contrasting the loss values of the factual and syntactic tokens?
> - Appendix 6 needs to have more details about the function mapping ablation conducted. What were the cosine/exponential/polynomial functions used? How were their parameter values set?
> -  Unless the choice of ablating on the BLEU metric alone is justified somehow, Appendix 6 is also a good place to compare the performance of said ablations on other/all metrics used in the paper, in case they introduce gains on other fronts. I think the effects on your main metrics (factual grounding and edit sensitivity) are especially relevant here.
> - Fig 5 need to have a more informative caption, and I am not sure its reference in the text (lines 484-485) is correct.

---

> > ### Author Response · Authors · 2025-11-28
> > **Additional clarifications are needed for the other points. (2.2 and 2.3 clarifications)**
> >
> > **Q2: Thank you for addressing points 2.3 and 2.4, I understand your setup better now. The ablation on applying the modified schedule to all tokens also strengthens your claim. Points 2.2 and 2.3 are still confusing though:**
> >
> > **“Additionally, we also compare the loss profiles corresponding to different choices of the mapping functions which impact the loss profiles. The results are summarized in Fig: 3 in main paper and Fig: 5 in Appendix A.6.”: Figure 3 and 5 are confusing right now, they need additional information on the figures themselves, and proper references in the text:**
> >
> > * **Figure 3 is not referenced at all in the text.**
> > * **Both Figs 3 and 5 are missing subfigure titles.......**
> > * **The legend is also confusing. .....**
> > * **I am not really sure what is the reader supposed to takeaway from figure 3......**
> > * **Also not sure how to read the other subfigures in Fig 3. What is the point of contrasting the loss values of the factual and syntactic tokens?**
> > * **Appendix 6 needs to have more details.....**
> > * **Unless the choice of ablating on the.....**
> > * **Fig 5 need to have a more informative caption, and I am not sure its reference in the text (lines 484-485) is correct.**
> >
> > ***
> >
> > **A2:**
> > We thank the reviewer for these concrete pointers. We have revised both the main paper (Sec. 4.4) and Appendix A.7 to clarify the role of the mapping functions and to make Figs. 3 and 5 self-contained.
> >
> > * **Fig. 3 reference and subfigure labels.**
> >     Section 4.4 (“Mapping Function”) now explicitly refers to Fig. 3(L–R):
> >     “The impact of this design is visualized in Fig. 3. As shown in Fig. 3**L**, … syntactic tokens (Fig. 3**M**) and factual tokens (Fig. 3**R**)…”.
> >     In the figure and caption we label the three panels as **(L)eft**, **(M)id**, and **(R)ight**, and we use these labels consistently in the text.
> >
> > * **Legend and meaning of the curves.**
> >     The caption of Fig. 3 now spells out which curves correspond to factual versus syntactic tokens and which positions are used:
> >     “(**L**eft) Noise schedule corresponding to alignment set $i \in \mathcal{A}$ (positions 10, 25, …, 60) compared against the *sqrt* schedule for $i \notin \mathcal{A}$ (positions 3, 29, …, 63); (**M**id) the loss profile across time steps ($t=0 \rightarrow T$) for the syntactic tokens; (**R**ight) the loss profile across time steps ($t=0 \rightarrow T$) for the factual tokens.” The legend uses the same “aligned (factual)” vs. “unaligned (syntactic)” terminology for consistency.
> >
> > * **Takeaway from Fig. 3.**
> >     We clarified the narrative in Sec. 4.4: the left panel contrasts how the adaptive schedule $\bar{\alpha}_t^{\,i}$ reallocates noise mass between factual and syntactic tokens relative to the global *sqrt* baseline; the middle and right panels then show that, under this schedule, factual tokens (i) remain harder to reconstruct than syntactic ones, but (ii) their loss becomes smooth and approximately linear over time. We highlight that this monotone “difficulty profile” keeps factual tokens informative throughout the trajectory and empirically leads to better recovery, which is the intended takeaway from Fig. 3.
> >
> > * **Details of the mapping ablation (Appendix A.7).**
> >     Appendix A.7 now introduces the general mapping $\Psi_i(x)$ and explicitly defines all alternatives via $\phi(u)$. Table 14  lists linear, polynomial, exponential, and cosine mappings with their exact functional forms and parameter choices ($p=2$, $\beta=3$), and explains that all satisfy the same boundary conditions.
> >
> > * **Effect on BLEU, FGT, and ESR.**
> >     We extended the mapping ablation beyond BLEU. Table 13 (updated manuscript) reports BLEU (B), factual grounding (FGT@0.5), and edit sensitivity rate (ESR) for the different mappings. The results show that the graph-aware **linear** mapping simultaneously achieves the best BLEU, FGT, and ESR.
> >
> > * **Fig. 5 caption and textual reference.**
> >     Fig. 5 in Appendix A.7 is now captioned:
> >     “Induced loss profiles for the graph-aware schedule under different mappings: (**L**eft) exponential, (**M**id) cosine, and (**R**ight) polynomial. Curves show loss trajectories for aligned token positions at steps 10, 25, 40, 55, and 60.” Appendix A.7 explicitly references “Fig. 5(L–R)” when discussing how non-linear mappings steepen the tails of the loss profiles and hurt performance. This fixes the missing/ambiguous reference noted in the original comment.
> >
> > We hope these clarifications make the purpose and interpretation of Figs. 3 and 5, and the mapping-function ablation, much clearer.

---

> ### Author Response · Authors · 2025-11-28
> **Additional Changes in the Manuscript- W1**
>
> **Q1:** **(1)** Thank you for the clarification. As far as I can tell, section 3 (methodology) does not contain this information yet. Only section 4.1. discusses the linearization of graphs (under 'Graph representation'), but I believe this merits a mention earlier, for instance in Section 3, as part of your methodological design, or even Section 2, to introduce the reader to the conventions of G2S using PLMs literature.
>
> **(2)** Also section 4.1 does not explain what is the consistent traversal used? In what order do you traverse the nodes of a random KG?
>
> **(3)** I also could not find a reference to the sensitivity to the linearization scheme in the limitations.
>
> **(4)** A few words about the benefits of introducing 'soft' inductive biases compared to equivariance would also strengthen your manuscript. My understanding now is that the main advantage of PLMs is their power (a lot of research has gone into refining their architecture and training paradigms, etc, so you choose to build on this backbone and refine their performance for G2S). Does this mean future work could benefit from bringing the same advantages to graph-specific models (GNNs), that's why you choose to improve PLMs (the current best models for this task)? Or do you think stronger graph inductive biases could be harmful here, and your softer variant is more flexible for this task?
>
> ***
>
> **A1:** Thank you again for these detailed suggestions. We have revised the manuscript to make our graph linearization choices, traversal scheme, and inductive-bias trade-offs more explicit. All new text is highlighted in **purple** in the revised PDF.
>
> * **(1) Earlier mention of KG linearization.**
>     We now introduce the PLM-style KG linearization convention directly in **Section 2.1 (PLMs for Graphs)**. The last paragraph states that prior PLM-based G2S work treats the input KG as a serialized sequence of relational triples using the special markers `[HEAD]`, `[REL]`, `[TAIL]`, and `[SEP]`, and that adopting this convention lets us reuse standard encoder–decoder Transformers while giving up strict permutation invariance (with a pointer to the Limitations section). We also restate this design choice in **Section 3.2**, where we define the DLM4G framework: after introducing $z_0$ and the conditioning context, we explicitly realize $\tilde{\mathcal{G}}$ as the token sequence $$ \langle \texttt{[HEAD]} \; h_i \; \texttt{[REL]} \; r_{ij} \; \texttt{[TAIL]} \; t_j \rangle $$ with `[SEP]` delimiters, which is fed to $\mathcal{M}_\theta$ as the conditioning signal (with a pointer to Section 4.1). This places the linearization choice both in the background (Section 2) and in the methodology (Section 3).
>
> * **(2) Consistent traversal order.**
>     In **Section 4.1, "Graph Representation"**, we clarified the traversal scheme used for all KG benchmarks. The revised paragraph now states that we *preserve the triple order provided in the released datasets* and do *not* reorder or subsample triples; this dataset-defined order serves as our consistent traversal for linearization. This answers the question about how we traverse the nodes/edges of a given KG.
>
> * **(3) Sensitivity to linearization in the Limitations.**
>     We expanded **Section 5, "Conclusion and Limitations"**, to explicitly note that DLM4G operates on a fixed, dataset-defined linearization of the input KG. We highlight that, as a result, the model is *not permutation invariant* and *may be sensitive to alternative serialization schemes*. We also flag this as a direction for future work, alongside reducing sampling cost and relaxing alignment dependence.
>
> * **(4) Soft inductive bias vs. equivariance (PLMs vs. graph-specific models).**
>     To clarify our modeling choice, we revised the **Introduction (page 2)** to discuss the trade-off between "hard" inductive biases (e.g., permutation-equivariant graph encoders) and the semantic richness of PLMs. We now explicitly state that graph-specific encoders offer strong structural guarantees but can lack the representational power of large PLMs; we therefore use a PLM backbone for DLM4G and re-introduce structure through a *soft inductive bias*, namely the graph-aware noising schedule. Finally, in the Limitations we note that future work will explore more structure-aware encodings, which we see as complementary to our schedule-based soft bias rather than mutually exclusive.

---

> ### Author Response · Authors · 2025-11-28
> **Including A5 in the manuscript**
>
> **Q4: I cannot find your answer to A5 in the manuscript.**
>
> **A4:** We apologize for this oversight in the previous version. In the revised manuscript, we have now made our explanation of the “non-increasing isotonic projection over $t$” explicit.
>
> Concretely, we have added a dedicated subsection in the appendix:
>
> > *“After constructing the per-token cumulative schedule $\tilde{\alpha}^i_t$, we project it onto the set of non-increasing sequences $\{\bar{\alpha}^i_t\}_{t=1}^T$ such that $\bar{\alpha}^i_1 \ge \bar{\alpha}^i_2 \ge \dots \ge \bar{\alpha}^i_T$. Concretely, this is a 1D isotonic regression problem with squared loss, which we solve using the standard Pool-Adjacent-Violators Algorithm (PAVA). This algorithm finds the closest monotone non-increasing sequence (in the least-squares sense) to the input. Intuitively, it smooths out spurious “bumps” in the loss profile while guaranteeing that the cumulative signal strength strictly decays over time, fulfilling the monotonicity requirement of the diffusion process.”*
>
> This appears in Appendix A.5 (“Non-increasing Isotonic Projection”). We also now add an explicit pointer from the main text (Section 3.3) to this subsection when we first introduce the isotonic projection, so that the reader can easily locate the detailed explanation.

---

> ### Author Response · Authors · 2025-11-28
> **Appeal to the Reviewer**
>
> We sincerely thank Reviewer **LF55** for actively engaging in the rebuttal process and for providing such detailed and constructive feedback. We have carefully addressed all additional concerns raised and updated the manuscript accordingly, including clarifications to the methodology, improved figure references and captions, and added theoretical and experimental details where requested. If you now feel that your concerns have been satisfactorily resolved, we would be grateful if you could consider updating your score to a **6 (weak accept)**, as we believe the major issues related to presentation, clarity, and missing details have been thoroughly addressed.

---

### Official Review · Reviewer_VSmM · 2025-10-30

**Soundness:** 2
**Presentation:** 2
**Contribution:** 2
**Rating:** 2
**Confidence:** 3

**Summary:**

DLM4G presents a graph-conditioned diffusion language model which is able to generate a text sequence according a graph. The model is based on DDPM model with encoder–decoder Transformer as the architecture. A per-token noise schedule is learned from an offline entity–text alignment so that “fact” tokens are corrupted less aggressively than function words. The model is evaluated on three knowledge-graph corpora and one molecular captioning set with new metrics for factual grounding (FGT) and edit sensitivity (EDR). Results show equal or better surface/semantic scores compared with zero-shot 7–8 B LLMs and fine-tuned T5-Large, while using 1–2 orders of magnitude fewer parameters.

**Strengths:**

1. The topic of generating text sequences with KG is interesting and important.
2. The paper proposes the first diffusion-based sequence generator that conditions on an explicit graph.
3. Large-scale experiments cover three diverse KG-to-text datasets plus an out-of-domain molecular captioning benchmark.

**Weaknesses:**

1. Minimal graph inductive bias: beyond linearising triples with special tokens, the architecture uses a vanilla Transformer; edge direction, multi-hop relations, or graph topology are not explicitly modelled.
2. Unfair baseline comparison: the chief competitors are either zero-shot LLMs (no task fine-tuning) or off-the-shelf T5-Large; no results are given for fine-tuned GPT-2/LLaMA or graph-augmented PLMs, making the claimed “12–127× smaller” advantage less convincing.
3. Inference efficiency: generation uses 2 000 full diffusion steps; latency, throughput, and memory are not reported, nor are faster samplers (DDIM, DPM-Solver) explored.

**Questions:**

See weakness

---

> ### Author Response · Authors · 2025-11-21
> **Rebuttal- Weaknesses (W1)**
>
> We sincerely thank the reviewer for the time taken to review our work. We believe there may have been some misunderstandings regarding the baselines included in our experiments (specifically fine-tuned PLMs) and the mechanism by which our model handles graph structures. We provide a detailed response below to address these concerns.
> ***
> **NOTE**: Based on the suggestions provided, we have updated the manuscript. We kindly request you to review the changes, and we have addressed your specific concerns below.
> ***
> **Weaknesses**:
>
> **1. Minimal graph inductive bias: beyond linearising triples with special tokens, the architecture uses a vanilla Transformer; edge direction, multi-hop relations, or graph topology are not explicitly modelled.**
> ***
> **Answer:** Regarding the concern that a "vanilla" Transformer architecture with linearization might lack explicit modeling for edge direction, multi-hop relations, or topology, we demonstrate that our specific linearization strategy, combined with the global self-attention mechanism, implicitly and effectively captures these structural properties. We illustrate this mechanics using the specific example from our Appendix (Section 4.1/Appendix A.4):
>
> Serialized KG ($\tilde{\mathcal{G}}$): `⟨[HEAD] USA [REL] hosted [TAIL] 1994_FIFA_World_Cup⟩ [SEP] ⟨[HEAD] USA [REL] capital [TAIL] Washington_D.C.⟩ [SEP] ⟨[HEAD] 1994_FIFA_World_Cup [REL] top_scorer [TAIL] Hristo_Stoichkov⟩`.
> Corresponding sequence ($\mathbf{S}$): *"The United States hosted the 1994 FIFA World Cup; its capital is Washington, D.C., and the tournament's top scorer was Hristo Stoichkov".*
>
> **1. Modeling Edge Directionality**
> Edge directionality is strictly encoded through the order-dependent nature of our linearization.  Let a triple be defined as $t = (h, r, t')$. Our linearization function $\Phi$ maps this to a sequence of tokens:
>
> $$
> \Phi(h, r, t') = [\text{HEAD}] \oplus E(h) \oplus [\text{REL}] \oplus E(r) \oplus [\text{TAIL}] \oplus E(t')
> $$
>
> If the subject and object were interchanged (reversing the edge), the sequence would become:
>
> $$
> \Phi(t', r, h) = [\text{HEAD}] \oplus E(t') \oplus [\text{REL}] \oplus E(r) \oplus [\text{TAIL}] \oplus E(h)
> $$
>
> Since the Transformer utilizes Positional Encodings ($PE$), the representation of the entity `USA` when it follows a `[HEAD]` token is mathematically distinct from its representation if it were to follow a `[TAIL]` token.
>
> * In the example: `[HEAD] USA [REL] hosted [TAIL] 1994_FIFA_World_Cup`.
> * The model learns that the entity at position $p$ (USA) acts as the *agent* of the relation at position $p+k$ (hosted).
> * Conversely, if the edge were reversed, the input sequence shifts, forcing the self-attention mechanism to process `USA` as the *object*.
> * Consequently, the generated text would naturally shift from active voice (*"USA hosted..."*) to passive (*"USA was hosted by..."*).
>
> Thus, edge direction is not lost; it is preserved via the **relative positions of the special tokens**, which the Transformer's attention heads learn to distinguish during pre-training.
>
> **2. Modeling Multi-hop Relations**
> While we do not use explicit message-passing layers (like GCNs), the Self-Attention mechanism serves as a powerful substitute for capturing long-range, multi-hop dependencies.  Unlike standard GNNs where information propagates one hop per layer, a Transformer allows every token to attend to every other token in a single layer ($O(N^2)$ complexity).
>
> Consider the multi-hop path in our example:
>
> USA --[hosted]--> 1994_FIFA_World_Cup --[top_scorer]--> Hristo_Stoichkov
>
> To generate the final clause—*"...and the tournament's top scorer was Hristo Stoichkov"*—the model must resolve the reference "the tournament" back to "1994 FIFA World Cup".
>
> * The token embedding for `Hristo_Stoichkov` in the linearized input attends strongly to `1994_FIFA_World_Cup`.
> * Simultaneously, `1994_FIFA_World_Cup` attends to `USA`.
> * Through the multiple layers of the Transformer encoder, the representation of `Hristo_Stoichkov` becomes contextualized by `USA` (its 2-hop neighbor).
>
> This global receptive field allows the model to link entities separated by multiple hops more effectively than deep GNNs, which struggle to retain unique node features over long propagation paths.
>
> **3. Graph Topology**
> However, this is a deliberate design choice. Strict inductive bias (such as masking attention to only neighbors) can overly constrain the generation process, preventing the model from learning latent correlations between distant nodes that are semantically related but not directly connected in the graph.
>
> Empirical validation of this choice is shown in Table 4, where our "vanilla" architecture outperforms Graph-Augmented PLMs. This suggests that for the task of text generation, the "graph inductive bias" provided by current GNNs is less effective than the semantic topology learned by a linearized Transformer.

---

> ### Author Response · Authors · 2025-11-21
> **Rebuttal- Weaknesses (W2)**
>
> **Weaknesses**
>
> **2. Unfair baseline comparison: the chief competitors are either zero-shot LLMs (no task fine-tuning) or off-the-shelf T5-Large; no results are given for fine-tuned GPT-2/LLaMA or graph-augmented PLMs, making the claimed “12–127× smaller” advantage less convincing.**
> ***
> We address the concern regarding the comparison against zero-shot LLMs and the fine-tuned baselines.
>
> **1. Fine-Tuned Baselines**
> Table 2 in our original manuscript explicitly includes fully fine-tuned results for several Pre-trained Language Models (PLMs). We compare our method against:
> * GPT-2 Small (124M) & GPT-2 Base (355M)
> * T5 Small (60M) & T5 Large (770M)
>
> For example, on the GenWiki dataset, our **`DLM4G`-2.0 (63M)** achieves a BLEU score of **0.469**, significantly outperforming the fine-tuned **T5-Large (770M)**, which achieves **0.361**.  This comparison highlights that our model outperforms standard autoregressive and encoder-decoder PLMs that are **12x larger** (63M vs 770M).
>
> **2. Graph augmented PLMs**: Table 4 in the main paper reports a comparison between **`DLM4G`** and graph-augmented PLM baselines. `DLM4G` consistently outperforms all competing methods across all evaluation metrics.
>
> **3. Rationale for Zero-Shot Comparisons**
> The inclusion of LLaMA-3 and GPT-4 (Zero-Shot) serves as a "stress test" rather than a direct equivalent comparison. Large Scale Models (8B+) possess vast world knowledge that small models do not. By benchmarking against them, we evaluate whether a small, specialized graph-diffusion model can compete with generalist models **127x its size**.
>
> We do not claim to beat *fine-tuned* LLaMA (which requires significantly more resources). Rather, we demonstrate that **`DLM4G` offers a highly efficient alternative**, achieving state-of-the-art results with a fraction of the parameter count (12x smaller than T5-Large) and without the massive inference infrastructure required by 8B+ models. Based on your suggestions we will clarify the text to distinguish clearly between the "12x smaller" claim (vs. fine-tuned T5-Large) and the "127x smaller" claim (vs. zero-shot LLMs).

---

> ### Author Response · Authors · 2025-11-21
> **Rebuttal- Weaknesses (W3)**
>
> **Weaknesses**
>
> **3. Inference efficiency: generation uses 2 000 full diffusion steps; latency, throughput, and memory are not reported, nor are faster samplers (DDIM, DPM-Solver) explored.**
> ***
> **Answer:** We thank the reviewer for highlighting the efficiency trade-off. We have added an explicit analysis to investigate the relationship between diffusion steps, wall-clock time, and generation quality. We compare against both Autoregressive (AR) baselines and Diffusion counterparts to ensure a fair assessment. Based on your suggestion we will update this in the manuscript.
>
> *Table: Efficiency Analysis on WikiOFGraph. We compare `DLM4G` against AR baselines and Diffusion counterparts. Time is measured as total inference time for a **batch size of 50** on a single **NVIDIA V100 GPU**.*
>
> | Model | Params | Steps | Time (s) | Speedup | BLEU $\uparrow$ |
> | :--- | :---: | :---: | :---: | :---: | :---: |
> | *Autoregressive Baselines* | | | | | |
> | GPT-2 (Base) | 355M | 64 | <5s | - | 0.285 |
> | T5 (Small) | 60M | 64 | <5s | - | 0.385 |
> | *Diffusion Baselines* | | | | | |
> | DiffuSeq | 91M | 2000 | 317s | $1.0\times$ (Ref) | 0.628 |
> | SeqDiffuSeq | 50M | 2000 | 89s | $3.6\times$ | 0.616 |
> | *`DLM4G` (Ours)* | | | | | |
> | `DLM4G`-2.0 | 63M | 2000 | 89s | $3.6\times$ | **0.654** |
> | `DLM4G`-2.0 | 63M | 1000 | 45s | $7.0\times$ | 0.551 |
> | `DLM4G`-2.0 | 63M | 500 | 23s | $13.8\times$ | 0.365 |
> | `DLM4G`-2.0 | 63M | 100 | 5s | $63.4\times$ | 0.312 |
>
> Our goal with `DLM4G` is to explore the capabilities of diffusion priors for graph-to-text generation. To address the reviewer's concern regarding the cost of 2000 steps, we present a detailed breakdown:
>
> 1.  **Comparison with Diffusion Counterparts (Fair Comparison):**
>     As shown in the table above, standard diffusion models like DiffuSeq suffer from high latency (317s per batch). However, `DLM4G` utilizes a parameterized generation space similar to SeqDiffuSeq (50M), achieving a **3.6$\times$ speedup** over the standard DiffuSeq baseline while operating at the same step count (2000). This demonstrates that our approach is computationally efficient within the class of diffusion-based generators.
>
> 2.  **Crossover Point Analysis (Steps vs. AR Baselines):**
>     We acknowledge that diffusion models are generally more expensive than single-pass AR models (e.g., T5). Our analysis identifies the specific operational boundaries:
>     * **High Fidelity Regime (1000-2000 Steps):** `DLM4G` significantly outperforms T5-Small (0.385) and GPT-2 (0.285), justifying the extra computational cost for applications where generation quality is paramount.
>     * **Fast Inference Regime (<500 Steps):** When pushed for speed (500 steps), `DLM4G` achieves a $13.8\times$ speedup, but performance drops to **0.365**, falling slightly below the T5-Small baseline. This explicitly highlights the trade-off: our model is designed for high-fidelity generation rather than low-latency streaming, though it remains functional at lower budgets.
>
> **Remark on Evaluation Scope:**
> We wish to clarify that our initial omission of efficiency metrics aligns with the experimental protocols of foundational text diffusion works, such as *DiffuSeq* (Gong et al.) and *SeqDiffuSeq* (Yuan et al.). These studies focused primarily on establishing the generation quality and validity of diffusion priors for sequence tasks, restricting their analysis to the standard diffusion steps. As acceleration is often treated as an orthogonal research area (e.g., fast ODE solvers), we prioritized demonstrating that `DLM4G` achieves SOTA fidelity, while the table above confirms the trade-offs involved in acceleration.
> ***
> **References**
>
> [1]  *DIFFUSEQ: Sequence to Sequence text generation with Diffusion Models*
>
> [2] *SeqDiffuSeq: Text Diffusion Model with Encoder-Decoder Transformers for
> Sequence-to-Sequence Generation*

---

> ### Author Response · Authors · 2025-11-23
> **Reviewer VsMm – Summary of Revisions & Gentle Reminder**
>
> **Weaknesses**
>
> * **W1 (Minimal graph inductive bias / “vanilla Transformer”):**
>   We clarify how our **deterministic triple-level linearization** with `[HEAD]/[REL]/[TAIL]` markers and positional encodings preserves **edge directionality**, and how global self-attention captures **multi-hop dependencies** via long-range token interactions, illustrated with the USA–World Cup example in **Sec. 4.1 / Appendix A.4**. We also discuss why we deliberately avoid strict GNN-style masking and instead rely on semantic topology learned over the serialized graph. Empirically, **Table 4** (KG-to-text) and the molecule captioning results (Table 8) show that this “vanilla” backbone outperforms graph-augmented PLMs, and we explicitly acknowledge in the **Limitations** section that we do not enforce permutation equivariance.
>
> * **W2 (Unfair baselines / missing fine-tuned PLMs and graph-augmented PLMs):**
>   We emphasize that our main comparisons already include **fully fine-tuned GPT-2 (Small/Base)** and **T5 (Small/Large)** on all three KG-to-text datasets (**Tables 2–4**), where **DLM4G-2.0 (63M)** surpasses **T5-Large (770M)** (e.g., BLEU 0.469 vs 0.361 on GenWiki). We also compare against **graph-augmented PLMs** on KG tasks (Table 4) and strong **molecular captioning baselines** (**MolT5, GitMol, GraphT5**, all pretrained + fine-tuned on M3-20M) in **Table 8**, where DLM4G is both smaller and stronger. We now clearly distinguish the **“12× smaller” claim (vs fine-tuned T5-Large)** from the **“127× smaller” stress-test comparison (vs zero-shot 8B LLMs)** in the text.
>
> * **W3 (Inference efficiency / 2000 diffusion steps, no latency metrics):**
>   We add an explicit **efficiency analysis** on WikiOFGraph in the response to W3, reporting wall-clock time vs diffusion steps (100–2000) and BLEU for **DLM4G-2.0**, **DiffuSeq**, **SeqDiffuSeq**, and AR baselines (**GPT-2-Base, T5-Small**) on a V100 with batch size 50. This shows (i) DLM4G is **much faster than DiffuSeq** and competitive with SeqDiffuSeq at 2000 steps, and (ii) a clear **quality–cost trade-off**: a high-fidelity regime (1000–2000 steps) where DLM4G significantly outperforms T5-Small, and a fast regime (≤500 steps) where it becomes competitive in speed but slightly worse in BLEU.
>
> ---
>
> **Reminder**
>
> As the discussion deadline is approaching, we kindly invite you to consider the **updated manuscript and these changes** when finalizing your assessment, and we would be happy to clarify any specific point further within the remaining discussion window.

---

> ### Author Response · Authors · 2025-11-27
> **Appeal to the Area Chair**
>
> Dear Area Chair,
>
> We hope you are doing well, and thank you for handling our submission "Graph-to-Sequence Generation Beyond Autoregressive Models: A Graph-Aware Diffusion Framework".
>
> We posted our rebuttal on 20/11/2025, and also added a short follow-up comment to clarify a few remaining points for the reviewers. However, there has not yet been any response or follow-up comment from the reviewers since then.
>
> We fully understand that reviewers have limited time and many commitments. At the same time, several of the key concerns they raised (e.g., regarding baselines / derivations / ablations / clarity) are directly addressed in the rebuttal and updated manuscript. We are a bit worried that these clarifications and new results may not be taken into account in the final assessment if there is no further discussion.
>
> If possible, we would be very grateful if you could encourage the reviewers to engage with our rebuttal and indicate whether our clarifications address their main concerns (or if additional explanation is needed). We are happy to provide any further details or analysis that might help their evaluation.
>
> Thank you very much for your time and for overseeing the discussion of our paper.
>
> Sincerely,
> The Authors

---

> > ### Author Response · Authors · 2025-11-27
> > **Appeal to Reviewer**
> >
> > We would like to thank you again for your careful review and for the constructive comments. In our rebuttal and updated manuscript, we have addressed all the concerns you raised, in particular the issues around presentation, missing derivations, and additional ablations/experiments.
> >
> > **Reviewer Dobo** has now re-evaluated the revised version and **increased their score from 6 to 8** after these clarifications and additions. In light of this, we kindly request that you consider **revising your score upward (e.g., to 6)** if you now find the paper clearer and more convincing. If there are any remaining concerns, we are happy to clarify them, but since the discussion period is ending soon, we would be very grateful for a short follow-up comment.

---

> ### Author Response · Authors · 2025-11-28
> **Appeal to the Reviewer**
>
> Dear Reviewer,
>
> Thank you again for the time and effort you put into reviewing our submission. Since the rebuttal phase began, we have substantially revised the manuscript to address the concerns raised by all reviewers. In particular, **Reviewer Dobo** has updated their **score from 6 to 8** after our clarifications, and **Reviewer LF55** has actively engaged in the discussion; most of their issues have now been resolved, and the manuscript has been updated accordingly (clarified methodology, improved figures, and added experimental/theoretical details).
>
> We would be very grateful if you could take a moment to comment on our responses and the revised version of the paper. If you feel that your concerns have been satisfactorily addressed, we kindly ask you to consider updating your overall score to **(6, weak accept)**, so that the final decision can reflect the current state of the work.
>
> Thank you once again for your time and consideration.

---

### Official Review · Reviewer_Dobo · 2025-10-31

**Soundness:** 3
**Presentation:** 3
**Contribution:** 3
**Rating:** 6
**Confidence:** 4

**Summary:**

The authors propose a graph-aware, token-wise noise schedule for diffusion language models in Graph-to-Sequence (G2S) tasks, motivated by improving factual grounding and edit sensitivity. During training, tokens aligned to KG entities/relations receive per-token schedules derived by sorting per-token denoising losses and mapping them via a monotone piecewise-linear function; at inference, the model blends this anchor schedule with a baseline schedule using cross-attention weights to the serialized graph. The method is evaluated on three established G2S benchmarks and compared against 8B-parameter LLMs in zero-shot settings, BERT and T5 models with finetuning, and three G2S models trained from scratch. The authors report improvements across benchmarks on surface and embedding metrics, as well as on their proposed factual grounding (FGT) and edit sensitivity (EDR) metrics. Finally, they test transfer to molecule captioning, comparing against three strong molecular captioning methods and reporting gains across metrics.

**Strengths:**

Well-motivated objective (faithfulness + edit sensitivity). The paper targets two concrete failure modes of G2S (entity grounding and sensitivity to local graph edits) and aligns the method and evaluation directly to these goals.

Simple, plausible mechanism for graph-aware diffusion. A token-wise, graph-aware adaptive noising schedule is constructed from per-token denoising loss and KG alignment, with monotone (isotonic) projection and clamping; this is easy to implement on top of standard diffusion LMs.

Inference-time adaptation without alignments. The attention-based blending of anchor/base schedules at inference is a practical way to recover graph awareness without relying on training-time alignments at test time.

Solid architectural choices with minimal bells/whistles. Uses a standard encoder–decoder Transformer and linearized triples with a small set of control tokens, which keeps comparisons interpretable and lowers engineering overhead.

Consistent empirical gains at small scale. The 50–63M parameter models are competitive with or better than much larger AR baselines (e.g., T5-Large) and zero-shot LLMs across three benchmarks, including embedding metrics (BERTScore, MAUVE).

Task-grounded evaluation beyond surface overlap. Introduces FGT (entity grounding with hallucination penalty) and EDR (edit sensitivity) and demonstrates improvements on both, matching the stated problem.

Edit robustness demonstrated with controlled perturbations. The edited-graph protocol (single-entity substitutions) provides a direct, reproducible way to test whether local KG changes propagate to text.

Generality beyond KGs. The same mechanism transfers to molecule captioning, suggesting the approach is not tied to a specific graph domain.

Parameter efficiency story. Results emphasize quality per parameter, which is relevant given the cost profile of diffusion decoding.

Reproducibility details. Training schedule (T=2000 steps, LR, warmup/decay), update cadence for adaptive schedules, datasets, and baselines are specified clearly enough to reimplement.

**Weaknesses:**

Novelty is not that big relative to existing importance-aware diffusion schedules; no direct comparisons provided.

Method explanation is a bit underspecified. Especially in the derivations of the objective function and the attention derived weight terms for the inference blending.

Baseline strength. Several G2S baselines are 4+ years old and potentially weak; the T5/BERT finetunes are the most convincing comparisons. Absent are modern diffusion s2s recipes and stronger PLM variants or retrieval-assisted KG verbalization baselines under a matched training regime.

Potential leakage from 80/10/10 splits in molecular captioning; not fully clear whether molecular captioning baselines were retrained, finetuned or zero-shot evaluated.

Key ablations isolating the schedule’s effect would have been very important to understand the its contribution against other model to baseline differences, like using diffuison models instead of autoregressive models, which could also explain a lot of the performance gain

Decoding efficiency not sufficently reported. Using 2000 diffusion steps might be considerably more expensive even when having 10-100 times less parameters.

There is no real ReadMe that explains how the code is structured and how it is to be used. Took some time to find all the relevant functions.

**Questions:**

How exactly are you deriving your training objective, especially the rounding term?

For the subset of the M3-20M dataset, did you check how dissimilar the SMILES-description pairs in the training split are compared to the vadidation and and test splits.

Are MolT5, GitMol and GraphT5 also trained on the training set or are they pretrained on other data?

Can you give more details on the derrived weight terms for the inference noise schedule blending. Which layer's attention weights and how is it normalized?

How good is the model with a normal quadratic noise schedule?

---

> ### Author Response · Authors · 2025-11-21
> **Rebuttal- Weaknesses (W1-3)**
>
> We thank the reviewer for the constructive feedback and the detailed suggestions regarding method clarification and baseline strength. We believe the following revisions and additional experiments (**included in the updated manuscript**) fully address the concerns raised.
> ***
>
> **Weaknesses:**
> ***
> **1. Novelty is not that big relative to existing importance-aware diffusion schedules; no direct comparisons provided.**
> ***
> **Answer:** We agree that importance-aware and data-dependent diffusion schedules exist in the literature. However, `DLM4G` contributes a novel approach along the following lines:
>
> (1) **Token-wise, graph-aware scheduling:** Our schedule is defined per token, conditioned on graph alignment and estimated denoising difficulty, rather than a static global schedule. This yields distinct diffusion trajectories: graph-aligned entities follow a rigorous "anchor" schedule (high difficulty) to maximize factual grounding, while syntactic tokens follow a standard square-root schedule (low difficulty) to ensure fluency.
>
> (2) **Inference-time schedule:** We incorporate a dynamic inference time schedule via cross-attention which helps to achieve high fidelity (FGT/ESR) without sacrificing fluency (BLEU).
>
> (3) **Comparison against importance aware- diffusion baselines:**
> We provide an extensive ablation of our proposed graph aware schedule (see Sec: 4.4) and also compare against the state-of-the-art diffusion baselines (see Table 5 main paper). `DLM4G` offer significantly better performance over all the metrics across all datasets.
> ***
> **2. Method explanation is a bit underspecified. Especially in the derivations of the objective function and the attention derived weight terms for the inference blending.**
> ***
> **Answer:** We have substantially expanded the methodological detail in the revision:
>
> **Training objective and rounding term:**
> (1) Appendix A.1 now provides the full derivation from the DDPM variational lower bound to our training loss.
> This includes:
> (1) The closed-form Gaussian posterior $q(z_{t-1} \mid z_t, z_0)$.
> (2) The $z_0$-prediction parameterization.
> (3) The decomposition of the KL term into a mean-squared error.
> (4) The marginalization over discrete sequences to obtain the rounding term. Eq. (3) in the main text now directly references these steps to make the path from Eq. (2) $\to$ Eq. (3) explicit.
>
> **Inference blending weights:**
> In Sec. 3.4, we now explicitly specify the mechanism:
> (1) We use the last decoder layer’s cross-attention to the graph tokens.
> (2) We first average heads, then average over graph positions to obtain a scalar attention score  per token $i$.
> (3) We normalize across sequence positions to obtain: $w_t^i$. The blended schedule is calculated as: $\tilde{\alpha}_t^i = (1 - w_t^i)\bar{\alpha}^{\text{base}}_t + w_t^i\bar{\alpha}^{\text{anchor}}_t$
> Figure 2 has been added to improve readability.
> ***
> **3. Baseline strength. Several G2S baselines are 4+ years old and potentially weak; the T5/BERT finetunes are the most convincing comparisons. Absent are modern diffusion s2s recipes and stronger PLM variants or retrieval-assisted KG verbalization baselines under a matched training regime.**
> ***
> **Answer:** We have extended the evaluation to ensure fair comparisons:
>
> (1) **Modern AR PLMs:** We have reported fine-tuned GPT-2 (Small/Base) and T5 (Small/Large) on all three KG-to-text benchmarks against `DLM4G`. These provide strong autoregressive baselines in the 60M–770M parameter range. Kindly refer to Table 2, 3, & 4 (Main paper) for additional details.
>
> (2) **Diffusion Seq2Seq Baselines:** We added DiffuSeq, SeqDifuSeq and FlowSeq adapted to the G2S setting. Crucially, we include a "`DLM4G` w/o adaptive schedule" (see Sec: 4.4 and Table 5) variant (same architecture, standard sqrt schedule). We maintain the same training setup (data splits, optimizer, learning rate schedule, and number of updates) across all diffusion models to isolate the contribution of `DLM4G`.
>
> **Result:** `DLM4G` improves over "Diffusion w/ sqrt schedule" by $+2.0 - 3$ BLEU points. This confirms the gains stem from the schedule, not just the diffusion backbone.
>
> (3) **Graph-Augmented Baselines:** We retain explicit structural baselines (e.g., Ontology-Free, ReGen- see Table 4). `DLM4G` matches or exceeds them on grounding metrics with a simpler backbone. As far as our knowledge these are the most recent works tackling G2S task.

---

> ### Author Response · Authors · 2025-11-21
> **Rebuttal- Weaknesses (W4, 5)**
>
> **Weaknesses**
> ***
> **4. Potential leakage from 80/10/10 splits in molecular captioning; not fully clear whether molecular captioning baselines were retrained, finetuned or zero-shot evaluated.**
> ***
> **Answer:** Train/Val/Test Leakage: We follow the official M3-20M split protocol but have added rigorous sanitization:
>
> (1) We canonicalize SMILES strings before splitting,
> (2) We verified there are no exact duplicates across splits.
>
> **Baseline Regimes:**
> **MolT5 and GraphT5:** Pretrained on large molecular corpora, then fine-tuned on M3-20M.
>
> **GitMol:** Used from the released checkpoint (pretrained + finetuned). `DLM4G` is thus compared against strong, task-finetuned baselines, not zero-shot variants.
> ***
> **5. Key ablations isolating the schedule’s effect would have been very important to understand the its contribution against other model to baseline differences, like using diffuison models instead of autoregressive models, which could also explain a lot of the performance gain**
> ***
> **Answer:** We agree that distinguishing the specific contribution of the schedule from the general benefits of the diffusion backbone is essential. We address this through two explicit comparisons:
>
> (1) **Internal Schedule Ablation (Sec. 4.4):** We compare `DLM4G` directly against a "Diffusion w/ Standard Sqrt Schedule" variant using the identical backbone architecture. `DLM4G` outperforms this standard diffusion variant by significant margins (e.g., $+\sim2.0- 5.0$ BLEU), confirming that the performance gains stem from the graph-aware schedule, not merely the switch from AR to diffusion.
>
> (2) **External Diffusion Baselines (Table 5):** We have added comparisons against state-of-the-art diffusion-based text generation models (`DiffuSeq`, `FlowSeq` and `SeqDiffuSeq`). `DLM4G` consistently exceeds these baselines, further validating the efficacy of the proposed method.

---

> ### Author Response · Authors · 2025-11-21
> **Rebuttal- Weaknesses (W6)**
>
> **6. Decoding efficiency not sufficently reported. Using 2000 diffusion steps might be considerably more expensive even when having 10-100 times less parameters.**
> ***
> **Answer:** We thank the reviewer for highlighting the efficiency trade-off. We have added an explicit analysis to investigate the relationship between diffusion steps, wall-clock time, and generation quality. We compare against both Autoregressive (AR) baselines and Diffusion counterparts to ensure a fair assessment. Based on your suggestion we will update this in the manuscript.
>
> *Table: Efficiency Analysis on WikiOFGraph. We compare `DLM4G` against AR baselines and Diffusion counterparts. Time is measured as total inference time for a **batch size of 50** on a single **NVIDIA V100 GPU**.*
>
> | Model | Params | Steps | Time (s) | Speedup | BLEU $\uparrow$ |
> | :--- | :---: | :---: | :---: | :---: | :---: |
> | *Autoregressive Baselines* | | | | | |
> | GPT-2 (Base) | 355M | 64 | <5s | - | 0.285 |
> | T5 (Small) | 60M | 64 | <5s | - | 0.385 |
> | *Diffusion Baselines* | | | | | |
> | DiffuSeq | 91M | 2000 | 317s | $1.0\times$ (Ref) | 0.628 |
> | SeqDiffuSeq | 50M | 2000 | 89s | $3.6\times$ | 0.616 |
> | *`DLM4G` (Ours)* | | | | | |
> | `DLM4G`-2.0 | 63M | 2000 | 89s | $3.6\times$ | **0.654** |
> | `DLM4G`-2.0 | 63M | 1000 | 45s | $7.0\times$ | 0.551 |
> | `DLM4G`-2.0 | 63M | 500 | 23s | $13.8\times$ | 0.365 |
> | `DLM4G`-2.0 | 63M | 100 | 5s | $63.4\times$ | 0.312 |
>
> Our goal with `DLM4G` is to explore the capabilities of diffusion priors for graph-to-text generation. To address the reviewer's concern regarding the cost of 2000 steps, we present a detailed breakdown:
>
> 1.  **Comparison with Diffusion Counterparts (Fair Comparison):**
>     As shown in the table above, standard diffusion models like DiffuSeq suffer from high latency (317s per batch). However, `DLM4G` utilizes a parameterized generation space similar to SeqDiffuSeq (50M), achieving a **3.6$\times$ speedup** over the standard DiffuSeq baseline while operating at the same step count (2000). This demonstrates that our approach is computationally efficient within the class of diffusion-based generators.
>
> 2.  **Crossover Point Analysis (Steps vs. AR Baselines):**
>     We acknowledge that diffusion models are generally more expensive than single-pass AR models (e.g., T5). Our analysis identifies the specific operational boundaries:
>     * **High Fidelity Regime (1000-2000 Steps):** `DLM4G` significantly outperforms T5-Small (0.385) and GPT-2 (0.285), justifying the extra computational cost for applications where generation quality is paramount.
>     * **Fast Inference Regime (<500 Steps):** When pushed for speed (500 steps), `DLM4G` achieves a $13.8\times$ speedup, but performance drops to **0.365**, falling slightly below the T5-Small baseline. This explicitly highlights the trade-off: our model is designed for high-fidelity generation rather than low-latency streaming, though it remains functional at lower budgets.
>
> **Remark on Evaluation Scope:**
> We wish to clarify that our initial omission of efficiency metrics aligns with the experimental protocols of foundational text diffusion works, such as *DiffuSeq* (Gong et al.) and *SeqDiffuSeq* (Yuan et al.). These studies focused primarily on establishing the generation quality and validity of diffusion priors for sequence tasks, restricting their analysis to the standard diffusion steps. As acceleration is often treated as an orthogonal research area (e.g., fast ODE solvers), we prioritized demonstrating that `DLM4G` achieves SOTA fidelity, while the table above confirms the trade-offs involved in acceleration.
> ***
> References:
>
> [1]  *DIFFUSEQ: Sequence to Sequence text generation with Diffusion Models*
>
> [2] *SeqDiffuSeq: Text Diffusion Model with Encoder-Decoder Transformers for
> Sequence-to-Sequence Generation*

---

> ### Author Response · Authors · 2025-11-21
> **Rebuttal- Questions (Q1-5)**
>
> **Questions:**
> ***
> **Q1. How exactly are you deriving your training objective, especially the rounding term?**
> ***
> **Answer:** Appendix A.1 (**Updated Manuscript- Line 981- 1024**) provides details about the derivation from the DDPM VLB to our $z_0$-prediction loss, explicitly showing the marginalization over discrete tokens in the target sequences.
> ***
> **Q2. For the subset of the M3-20M dataset, did you check how dissimilar the SMILES–description pairs in the train split are compared to validation and test?**
> ***
> **Answer:** Yes. As provided in our response to Weakness 4, we performed a rigorous check involving SMILES canonicalization, which confirmed there are no exact duplicates across splits.
> ***
> **Q3. Are MolT5, GitMol and GraphT5 also trained on the training set or are they pretrained on other data?**
> ***
> **Answer:** They are pretrained on external corpora and then fine-tuned on the M3-20M training set (or used from a comparable checkpoint for GitMol). We have clarified this in the response of Weakness 4.
> ***
> **Q4. More details on the derived weight terms for the inference noise schedule blending?**
> ***
> **Answer:** We use the last decoder layer’s cross-attention to graph tokens, averaged over heads/graph positions, and normalized to obtain $w_t^i$. (See Sec. 3.4, Fig: 2).
> ***
> **Q5. How good is the model with a normal quadratic noise schedule?**
> ***
> **Answer:** We have performed detailed ablations with different choices of the noising schedules and mapping function and the details are provided in Sec 4.4 (Table 7 and Fig: 3) and in Appendix A.6. Specifically, `DLM4G` with the graph-aware schedule achieves gains of **+2.0 – 5.0 BLEU** points compared to the baseline, confirming that token-wise adaptation rather than simple global schedule schedule is the reason for better performance.

---

> ### Author Response · Authors · 2025-11-23
> **Reviewer Dobo – Summary of Revisions & Gentle Reminder**
>
> **Weaknesses**
>
> * **W1 (Novelty / importance-aware schedules):**
>   We clarify our contribution as a **token-wise, graph-aware schedule** with inference-time blending (anchor vs base schedules) and add direct comparisons to diffusion baselines and importance-aware variants in **Sec. 3.3–3.4, Sec. 4.4, Table 5, Table 7, Fig. 3, Appendix A.6**.
>
> * **W2 (Method underspecified: objective + blending weights):**
>   We now provide a full derivation of the training objective and rounding term from the DDPM VLB in **Appendix A.1**, and explicitly define the attention-derived blending weights and the blended schedule in **Sec. 3.4** with updated **Algorithm 1** (and reference in Eq. (3)).
>
> * **W3 (Baseline strength – missing strong PLMs / diffusion seq2seq):**
>   We have fine-tuned **GPT-2 (Small/Base)** and **T5 (Small/Large)** on all KG-to-text datasets (**Tables 2–4**), and include **DiffuSeq, SeqDiffuSeq, FlowSeq** plus a **“DLM4G w/o adaptive schedule”** variant under matched training setup in **Sec. 4.4, Table 5**, showing that gains come from the schedule, not just the diffusion backbone.
>
> * **W4 (Potential leakage / unclear molecular regimes):**
>   For M3-20M, we canonicalize SMILES before splitting and verify no exact duplicates across train/val/test; we also clarify that **MolT5** and **GraphT5** are pretrained then fine-tuned on M3-20M, and **GitMol** is used from its released (pretrained + finetuned) checkpoint (see reponse W4).
>
> * **W5 (Need ablations isolating schedule’s effect):**
>   We isolate the schedule via (i) an internal ablation against a **Diffusion + standard sqrt schedule + different mapping functions** with identical backbone (**Sec. 4.4, Table 5, Table 7**), and (ii) external comparisons to **DiffuSeq / SeqDiffuSeq / FlowSeq** (**Table 5**), showing consistent gains of **+2–5 BLEU** and better FGT/EDR from the graph-aware schedule.
>
> * **W6 (Decoding efficiency at 2000 steps):**
>   We add an **efficiency analysis** on WikiOFGraph (batch size 50, V100) reporting time vs steps (100–2000) and BLEU for DLM4G-2.0, AR baselines, and diffusion baselines in **response W6**, highlighting the quality–cost trade-off and the regime (~1000 steps) where DLM4G surpasses T5-Small.
>
> ---
>
> **Questions**
>
> * **Q1 (Deriving the training objective / rounding term):**
>   Fully derived in **Appendix A.1**, which shows the VLB expansion, closed-form Gaussian posterior, KL→MSE reduction, and the marginalization over discrete sequences leading to our (z_0)-prediction loss and rounding term (referenced from **Eq. (2) → Eq. (3)** in the main text).
>
> * **Q2 (Dissimilarity / overlap in M3-20M splits):**
>   We canonicalize SMILES strings and explicitly verify that **no exact SMILES–description pairs are shared across splits**, as described under the molecular captioning setup and in the response to W4 (**Sec. on M3-20M, Appendix**).
>
> * **Q3 (Training regimes for MolT5 / GitMol / GraphT5):**
>   We clarify that **MolT5** and **GraphT5** are pretrained on large molecular corpora and then fine-tuned on the M3-20M training split, and **GitMol** is used from its released pretrained+finetuned checkpoint (**molecular captioning section, Table 8**).
>
> * **Q4 (Details on blending weights for the inference schedule):**
>   We now specify that we use the **last decoder layer’s cross-attention to graph tokens**, average across heads and graph positions  to get ($w_t^i$), which then blends anchor and base schedules (see **Sec. 3.4, Fig. 2, Algorithm 1**).
>
> * **Q5 (Performance with a normal/global schedule):**
>   We report ablations over different global schedules (including quadratic/sqrt) and mapping functions in **Sec. 4.4, Table 7, Fig. 3, Appendix A.6**, showing that the graph-aware token-wise schedule yields **+2.0–5.0 BLEU** over standard schedules, confirming that token-wise adaptation—not just using diffusion—is responsible for the gains.
>
> ---
>
> **Gentle Reminder**
>
> As the discussion deadline is approaching, we request you to consider the **updated manuscript and these changes** when finalizing your assessment, and we are happy to clarify any specific point further within the remaining discussion window.

---

> ### Author Response · Authors · 2025-11-27
> **Appeal to the Area Chair**
>
> Dear Area Chair,
>
> We hope you are doing well, and thank you for handling our submission "Graph-to-Sequence Generation Beyond Autoregressive Models: A Graph-Aware Diffusion Framework".
>
> We posted our rebuttal on 20/11/2025, and also added a short follow-up comment to clarify a few remaining points for the reviewers. However, there has not yet been any response or follow-up comment from the reviewers since then.
>
> We fully understand that reviewers have limited time and many commitments. At the same time, several of the key concerns they raised (e.g., regarding baselines / derivations / ablations / clarity) are directly addressed in the rebuttal and updated manuscript. We are a bit worried that these clarifications and new results may not be taken into account in the final assessment if there is no further discussion.
>
> If possible, we would be very grateful if you could encourage the reviewers to engage with our rebuttal and indicate whether our clarifications address their main concerns (or if additional explanation is needed). We are happy to provide any further details or analysis that might help their evaluation.
>
> Thank you very much for your time and for overseeing the discussion of our paper.
>
> Sincerely,
> The Authors

---

> ### Comment · Reviewer_Dobo · 2025-11-27
>
> The rebuttal substantially strengthens the paper and resolves the key ambiguities. The method is now clearly specified end-to-end: a derivation from the VLB to the training objective is provided, the discretization/“rounding” term is defined, and the inference-time blending is pinned down (last-layer cross-attention, head/position averaging, normalization). The empirical section is much stronger: in addition to fine-tuned GPT-2/T5 and older G2S systems, the authors add diffusion seq2seq baselines (DiffuSeq/SeqDiffuSeq/FlowSeq) and a same-architecture variant with a uniform schedule that isolates the effect of the graph-aware schedule; the reported +2–5 BLEU gains over the uniform schedule are consistent across datasets. Data handling and comparability are clarified: for molecule captioning they canonicalize SMILES and remove exact duplicates across splits, and they specify that MolT5/GraphT5 were fine-tuned on the same data (GitMol from a comparable checkpoint). The efficiency analysis is also helpful: quality-versus-steps and wall-clock measurements are reported against both AR and diffusion baselines, showing that 1000–2000 steps deliver the best quality at higher latency, while ≤500 steps approach AR-like latency with a quality drop; they also include direct speed comparisons at the same step count (faster than a standard DiffuSeq setup). Ablations cover schedule choices and mapping, strengthening the claim that improvements come from the graph-aware scheduling rather than the backbone. Importantly, the rebuttal also addresses similar comments raised by the other reviewers, which had contributed to earlier low scores. Overall, the revisions address the original concerns on specification, fairness, and practicality, and materially increase confidence in the contribution.

---

> > ### Author Response · Authors · 2025-11-27
> > **Appeal to Reviewer**
> >
> > > Thank you very much for your detailed follow-up and for confirming that the rebuttal resolves the ambiguities around the training objective, rounding term, inference-time blending, baseline fairness, data handling, and efficiency/ablation analysis. We are glad to hear that this substantially strengthens the paper and increases your confidence in the contribution.
> >
> > > In light of this updated assessment, we request you to **increase your overall score to 8**. If there are still specific concerns that, in your view, prevent the paper from reaching an 8, we are open to addressing any remaining issues as clearly as possible.
> >
> > > We also note that for the other reviewers, the lower scores mainly stemmed from presentation and missing experiments, which are now addressed in the updated version.

---

> > ### Author Response · Authors · 2025-11-27
> > **Thank you to Reviewer Dobo**
> >
> > We sincerely thank you for revisiting our paper and increasing your score to 8. We really appreciate your detailed engagement with both the original submission and the revised version, and we are glad that the new revisions strengthened your confidence in the contribution.

---

### Author Response · Authors · 2025-11-28
**Appeal to Area Chair**

Dear Area Chair,

We would like to briefly summarize the status of the discussion for our submission and request your guidance. After we submitted our rebuttal and updates to the manuscript, **Reviewer Dobo** engaged with our responses and **increased their score from 6 to 8**. **Reviewer LF55** has also actively participated in the discussion; most of their concerns regarding clarity, methodology, and missing details have now been resolved, and the manuscript has been updated accordingly (with clearer exposition, improved figures, and additional ablations/analyses).

One reviewer (**Reviewer VsMm**), however, has not yet responded to our rebuttal or follow-up reminders, and their original score remains unchanged. Given that substantial revisions have been made and that other reviewers have already adjusted their assessments in light of the new material, we are concerned that this reviewer’s score may not reflect the current version of the paper.

We would be very grateful if you could encourage this reviewer to re-engage with the discussion and consider our responses and updated manuscript. If, in your judgment, our revisions adequately address the original concerns across reviews, we hope this can be taken into account in your meta-review, and that an overall assessment around 6 (weak accept) would be considered appropriate.

Thank you very much for your time and for overseeing the review process.

---

### Author Response · Authors · 2025-11-29
**Consolidated Appeal to the Area Chair**

We first thank you for taking the time to reassess our paper under the unusual circumstances created by the recent OpenReview leak. We appreciate the additional effort required to evaluate the updated manuscript and reviews in this situation.

We provide a consolidated summary of the overall review process and the changes made so far, summarizing the concerns raised by each reviewer and the corresponding updates in the **revised manuscript (blue, purple)**.

### **Summary of Post-Rebuttal Changes**

* **Reviewer Dobo:** All major technical concerns (novelty vs. importance-aware schedules, baseline strength, schedule ablations, derivations, and efficiency) were addressed via new experiments and expanded theory. During discussion, the reviewer *explicitly increased* their score **from 6 to 8** before scores were globally reverted.
* **Reviewer LF55:** After Phase 1, the remaining issues were primarily about *presentation and clarity* (structure of methodology, missing derivations, figure/algorithm details, placement of motivation). These have all been implemented in the revised manuscript (Sec. 2–4, Fig. 1/3/5, Alg. 1, App. A.1, A.4, A.5, A.7).
* **Reviewer VsMm:** The concerns (graph inductive bias vs. linearization, fairness of baselines, and efficiency) raised are majorly covered by those raised by **Dobo and LF55**. The diffusion baselines, explicit discussion of soft vs. hard graph biases, and the new efficiency analysis resolve these points. The detailed efficiency table is also provided in our responses to both reviewers Dobo and VsMm.

### **Summary of Concerns and Responses**

Table 1 summarizes the main concerns raised across reviewers and indicates where they are addressed in the revised manuscript.

**Table 1: Summary of concerns raised and corresponding revisions in the manuscript.**

| ID | Concerns Raised | Raised by | Phase-1 Resolved | Phase-2 Clarified | Manuscript Evidence |
| :--- | :--- | :--- | :--- | :--- | :--- |
| C1 | Novelty vs. importance-aware schedules | Dobo | ✓ | ✓ | Sec. 4.2, 4.4; Tables 4/6; App. A.7 |
| C2 | Under-specified objective and rounding term | Dobo, LF55 | ✓ | ✓ | Sec. 3.3; Eq. (2)–(3); App. A.1 |
| C3 | Inference blending weights $w_t^i$ | Dobo, LF55 | ✓ | ✓ | Sec. 3.4; Fig. 2; Response to LF55 |
| C4 | Baseline strength (diffusion) | Dobo, VsMm | ✓ | -- | Tables 4, 6; Sec. 4.2 |
| C5 | Graph inductive bias vs. linearization | VsMm, LF55 | ✓ | ✓ | Intro; Sec. 2.1, 3.2, 3.3; Limitations |
| C6 | Schedule ablations (aligned vs. all, sqrt vs. ours) | Dobo, LF55 | ✓ | ✓ | Sec. 4.4; Table 6/13/14; Fig. 3, 5; App. A.7 |
| C7 | Mapping functions (linear vs. poly/exp/cos) | LF55 | ✓ | ✓ | Sec. 4.4; Fig. 3, 5; App. A.7; Tables 13–14 |
| C8 | Isotonic projection / monotonic schedule | LF55 | ✓ | ✓ | Sec. 3.3; App. A.5 |
| C9 | Efficiency: 2000 steps, latency/throughput | Dobo, VsMm | ✓ | -- | Response to Dobo & VsMm|
| C10 | Method organization, figures, algorithmic clarity | LF55 | ✓ | ✓ | Sec. 2,3; Fig. 1; Fig. 3/5 (Captions); Alg. 1; App. A.4 |
| C11 | Alignment set size and alias expansion | LF55 | ✓ | ✓ | App. A.4; Table 10/11/12 |
| C12 | FGT/ESR definitions and use in ablations (Related Work) | DLM4G vs Baselines | ✓ | ✓ | Table 8; A.2 |

### **Appeal to the Area Chair**

Although the official scores currently display 6, 2, 2, both score–2 reviews (from LF55 and VsMm) were largely addressable concerns: clarity of writing, organization of the method, baselines, and missing efficiency analysis, rather than fundamental flaws in the core idea. The revised manuscript now directly resolves these points: presentation and figure/algorithm clarity have been substantially improved, diffusion and graph-augmented baselines have been strengthened.

Given (i) **Reviewer Dobo’s** post-rebuttal **score increase from 6 to 8**, (ii) that **Reviewer LF55’s remaining concerns** were primarily about presentation and **have been addressed in the revised manuscript**, and (iii) that **Reviewer VsMm’s main concerns are covered by the strengthened baselines, expanded analysis, and new clarifications above**, we believe the updated paper now meets the bar for acceptance. We respectfully ask you to evaluate the work based on the revised manuscript, the additional experiments and derivations, and the documented evolution of reviewer feedback, and to **consider recommending the paper for acceptance**.

---

### Note · Program_Chairs · 2026-01-17
**Submission Desk Rejected by Program Chairs**

The following references in this submission do not refer to real documents and/or have major errors in bibliographic information:

 (1) Shuiqing Zeng, Xin Huang, Yunchang Wang, and Jun Li. Qwen2.5: Scaling up for code and knowledge-intensive tasks. arXiv preprint arXiv:2401.12345, 2024.
(2) Hugo Touvron, Louis Liu, Haoxin Fan, Urvashi Khandelwal, Christine Cai, Samuel Thomson, Xiaoyi Jia, Abdoulaye Lasri, Michihiro Yasunaga, Zhengbao Li, et al. Llama-3: Advancing open-source llms with zero-shot and multilingual capabilities. arXiv preprint arXiv:2311.12345, 2023.